# AdaptGrad: Adaptive Sampling to Reduce Noise

**Linjiang Zhou**[1]    **Chao Ma**[2]    **Zepeng Wang**[2]    **Libing Wu**[2]    **Xiaochuan Shi**[2]*

[1]Key Laboratory of Aerospace Information Security and Trusted Computing, Ministry of Education
School of Cyber Science and Engineering, Wuhan University, Wuhan 430072, China
[2]School of Cyber Science and Engineering, Wuhan University, Wuhan 430072, China
{linjiang, chaoma, wangzepeng, wu, shixiaochuan}@whu.edu.cn

## Abstract

Gradient smoothing is an efficient approach to reducing noise in gradient-based model explanation methods. SmoothGrad adds Gaussian noise to mitigate much of this noise. However, the crucial hyperparameter in this method, the variance $\sigma$ of the Gaussian noise, is often set manually or determined using a heuristic approach. This results in the smoothed gradients containing extra noise introduced by the smoothing process. In this paper, we aim to analyze the noise and its connection to the out-of-range sampling in the smoothing process of SmoothGrad. Based on this insight, we propose AdaptGrad, an adaptive gradient smoothing method that controls out-of-range sampling to minimize noise. Comprehensive experiments, both qualitative and quantitative, demonstrate that AdaptGrad could effectively reduce almost all the noise in vanilla gradients compared to baseline methods. AdaptGrad is simple and universal, making it a practical solution to enhance gradient-based interpretability methods to achieve clearer visualization. All code would be found in `https://github.com/AiShare-WHU/AdaptGrad`.

## 1   Introduction

Explanation of the deep learning model is a critical part of applications of artificial intelligence (AI) with human interaction. For example, explanation methods are crucial in these data-sensitive and decision-sensitive fields such as medical image analysis [4], financial data analysis [55] and autonomous driving [1]. Additionally, given the prevalence of personal data protection laws in most countries and regions, fully black-box AI models may face intense legal scrutiny [14].

In recent years, some explanation methods have attempted to explain neural network decisions by visualizing the decision rationale and feature importance [31]. These local explanation techniques aim to provide explanations for individual samples. Moreover, the explanation process of these methods often leverages the gradients of the neural network. For example, Grad-CAM [35], Grad-CAM++ [11], and Score-CAM [46] use gradients to generate the weights of class activation maps.

Gradients of input samples are critical information for analyzing deep neural networks [41]. However, these gradients often contain a significant amount of noise, primarily due to the complex structure and numerous parameters in neural networks [30]. As highlighted in [2, 25, 51], this noise can significantly affect the ability of explanation methods to extract latent learning features. Furthermore, caused by both local noise and gradient saturation, sample gradients may fail to accurately explain the influence of input values on model decisions [42]. Therefore, reducing the noise in gradients is an important step toward improving the interpretability of deep learning models.

---

*Linjiang Zhou and Chao Ma contribute equally to this work. Xiaochuan Shi is the corresponding author of this paper.

39th Conference on Neural Information Processing Systems (NeurIPS 2025).

SmoothGrad [40] is currently the most widely applied and empirically proven simplest yet effective method for gradient smoothing. Although other methods, such as NoiseGrad [9], have also been proposed for gradient smoothing, SmoothGrad remains the most widely used due to its universality and practicality. However, as mentioned in [5, 32, 9, 31], the underlying principles of SmoothGrad have not been thoroughly explored. Additionally, its key parameter $\sigma$, the variance of Gaussian noise, is typically set empirically. We found that this setup causes SmoothGrad to introduce additional noise during its sampling process, which consequently leads to the smoothed gradient still retaining a significant amount of noise.

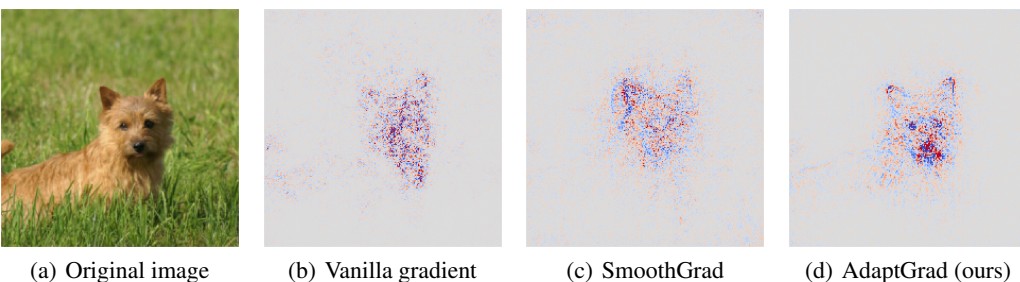

| (a) Original image | (b) Vanilla gradient | (c) SmoothGrad | (d) AdaptGrad (ours) |

Figure 1: An example to compare the visual performance between different gradient smoothing methods.

In this paper, we rethink the noise sources in SmoothGrad utilizing the convolution formula. We discover the relationship between the out-of-range sampling behavior caused by its key hyperparameter settings and the noise introduced during the sampling process. This discovery enables us to theoretically analyze the shortcomings of SmoothGrad and subsequently design an adaptive gradient smoothing method, AdaptGrad. Figure 1 illustrates an example of AdaptGrad. We not only theoretically prove that AdaptGrad outperforms SmoothGrad, but also demonstrate that our AdaptGrad is capable of eliminating almost all the noise while the smoothed gradients reveal richer detailed features. Similarly to SmoothGrad, AdaptGrad is also simple and universal, making it applicable for improving gradient-based interpretability methods. In addition, we comprehensively demonstrate the superiority of AdaptGrad through experiments with other gradient-based interpretability methods.

## 2 Related Work

In general, a neural network can be considered as a function $F(\mathbf{x}; \theta) : \mathbb{R}^D \to \mathbb{R}^C$ with the trainable parameters $\theta$ and its output could be probability, logit, etc. Here, $D$ is the input dimension of the neural network, and $C$ is the output dimension. In the example of a classification function, the neural network will output a score for each class $c$, where $c \in \{1, \cdots, C\}$. To simplify the analysis, we can focus on the output of the neural network for a single class $c$, and the neural network can be viewed as a function $F^c(\mathbf{x}; \theta) : \mathbb{R}^D \to \mathbb{R}$, which maps the input $\mathbb{R}^D$ to the 1-dimensional $\mathbb{R}$ space. For simplicity, we use $F(\mathbf{x})$ to represent the neural network and only consider the output of the neural network on a single class $c$.

The gradient of $F(\mathbf{x})$ could be presented as Equation 1.

$$G(\mathbf{x}) = \frac{\partial F(\mathbf{x})}{\partial \mathbf{x}}. \tag{1}$$

A possible explanation [30] for the large amount of noise in the original gradient is that, considering the local surroundings of a sample, neural networks tend not to present an ideal linearity but rather have a very rough and nonlinear decision boundary. So the complexity of neural networks and the input features usually leads to the unreliability of the vanilla sample gradients.

Similarly, gradient maps such as those in Figure 1 are commonly referred to as saliency maps or heatmaps. For simplicity, this paper uniformly refers to them as saliency maps and also refers to the results generated by other explanation methods as saliency maps. The highlighted areas in these maps indicate the relevant features learned by the neural network or the basis for the decisions.

The methods for reducing gradient noise can be roughly categorized into the following two categories:

**Adding noise to reduce noise.** SmoothGrad proposed by [40] introduces randomness to smooth the noisy gradients. SmoothGrad averages the gradients of random samples in the neighborhood of the input $\mathbf{x}_0$. This could be formulated as shown in Equation 2.

$$G_{sg}(\mathbf{x}) = \frac{1}{N} \sum_i^N G(\mathbf{x} + \varepsilon), \text{where } \varepsilon \sim \mathcal{N}^D(0, \Sigma_{sg}), \Sigma_{sg} = I_D * \sigma^2 \tag{2}$$

In Equation 2, $N$ is the sample times, and $\varepsilon$ is distributed over $D$-dimension $\mathcal{N}^D(0, \Sigma_{sg})$. Similarly to SmoothGrad, NoiseGrad, and FusionGrad presented in [9], additionally add perturbations to the model parameters. And FusionGrad is a mixup of NoiseGrad and SmoothGrad. These simple methods are experimentally verified to be efficient and robust [13].

**Improving backpropagation to reduce noise.** Deconvolution [53] and Guided Backpropagation [41] directly modify the gradient computation algorithm of the ReLU function. Integrate Gradient (IG) [42, 23, 24, 49, 52] was proposed to replace the original gradient for interpretation and was shown to have axiomatic completeness. Some other methods such as Feature Inversion [15], Layerwise Relevance Propagation [7], DeepLift [36], Occlusion [5], DeepTaylor [30] employ some additional features to approximate or improve the gradient for precise visualization.

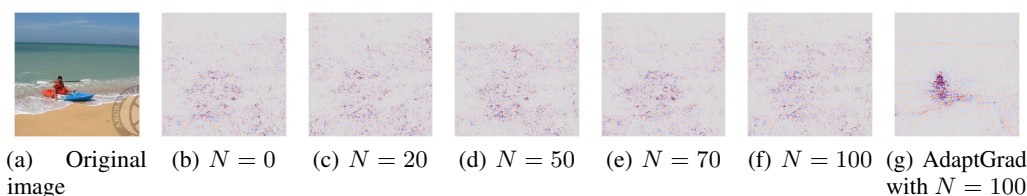

(a) Original image    (b) $N = 0$    (c) $N = 20$    (d) $N = 50$    (e) $N = 70$    (f) $N = 100$    (g) AdaptGrad with $N = 100$

Figure 2: The visual saliency map $G_{sg}$ of SmoothGrad with different sampling number $N$ and $\alpha = 0.2$. The classification model is VGG16 [37], and this image is from ILSVRC2012 [26].

## 3 Convolution for Smoothing

SmoothGrad is a simple yet effective gradient smoothing method. However, some of its underlying principles have not been fully discussed. As mentioned in [9], SmoothGrad is essentially a Monte Carlo method. Thus, we could further derive the definition of SmoothGrad to gain a deeper understanding of noise in gradient smoothing.

### 3.1 Monte Carlo Approximation for Convolution

In Section 2, SmoothGrad is formulated in Equation 2. As summarized in previous work [47], SmoothGrad is essentially a type of convolution. In the form of Monte Carlo integration, SmoothGrad could be redefined as Equation 3.

$$G_{sg}(\mathbf{x}) = \frac{1}{N} \sum_i^N G(\mathbf{x} + \varepsilon) = \frac{1}{N} \sum_i^N \frac{G(\mathbf{x} + \varepsilon)\varphi(\varepsilon)}{p(\varepsilon)}, \text{where } \varphi(\cdot) = p(\cdot) \tag{3}$$

In Equation 3, the $p(\cdot)$ is the Probability Density Function (PDF) of $D$-dimensional distribution $\mathcal{N}^D(0, \Sigma_{sg})$. And according to the Monte Carlo integration principle, the limit of $G_{sg}$ can be estimated as $N$ approaches infinity (sampling an infinite number of times). So we could obtain the upper limit of $G_{sg}$ in Equation 4.

$$\lim_{N \to \infty} G_{sg}(\mathbf{x}) = \lim_{N \to \infty} \frac{1}{N} \sum_i^N \frac{G(\mathbf{x} + \varepsilon)\varphi(\varepsilon)}{p(\varepsilon)} = \int G(\mathbf{x} + \varepsilon)\varphi(\varepsilon)d\varepsilon = (G * \varphi)(\mathbf{x}) \tag{4}$$

In Equation 4, $*$ is the convolution operator. From Equation 4, we could observe that the function $p(\cdot)$ (equal to $\varphi(\cdot)$) acts as both a convolution kernel $\varphi(\cdot)$ and a sampling distribution $p(\cdot)$. The reason why $\varphi(\cdot)$ and $p(\cdot)$ must be equal arises from computational considerations, as the PDF values of

high-dimensional distributions are typically very small, which can lead to floating-point underflow. For ease of distinction, in the following text, we will use $p(\cdot)$ to denote both the sampling distribution and the convolution kernel. So we extend the definition of SmoothGrad as shown in Equation 5.

$$G_{sg}(\mathbf{x}) \simeq (G * p)(\mathbf{x}), \ p = PDF(\mathcal{N}^D(0, \Sigma_{sg})) \tag{5}$$

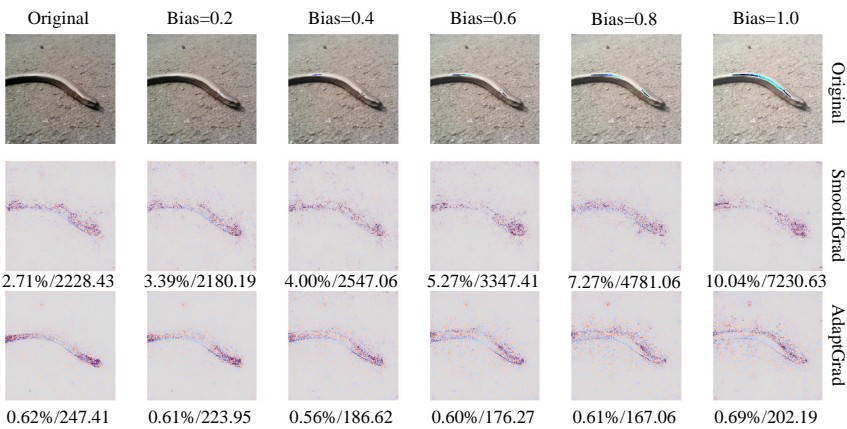

Figure 3: An example of the relationship between out-of-bound sampling behavior and extra noise. By adding a bias to the input image (illustrated as Bias in the figure), the out-of-bound sampling behavior of SmoothGrad increases progressively (the proportion of out-of-bound pixels / the value of out-of-bound and labeled at the bottom of the saliency map).

## 3.2  Noise in the Smoothed Gradients

As implied by Equation 5, when the sampling number $N$ is sufficiently large, $G_{sg}$ has an upper limit. In Figure 2, a visual example of SmoothGrad, as $N$ increases, around 50 to 70, $G_{sg}$ gradually converges. However, even as $G_{sg}$ approaches the limit, the residual noise in $G_{sg}$ could still affect the details in the saliency map.

Based on our findings in Section 3.1, SmoothGrad can be understood as a convolution of gradient functions. The convolution of the Gaussian kernel cannot completely remove all noise. Therefore, there will inevitably be some inherent noise in the smoothed gradient. However, we find that in the SmoothGrad method, there is also new noise caused by the sampling range, as this part of the noise is generated by the SmoothGrad method itself, so we call it **extra noise**. Next, we will analyze and prove the existence of extra noise.

The $\sigma$, a key parameter in SmoothGrad, is the variance of sample distribution $N^D(0, \Sigma_{sg}), \Sigma_{sg} = I_D * \sigma^2$. SmoothGrad employs a simple strategy to select an appropriate value for $\sigma$. Assume that the minimum value of the input data is denoted as $x_{min}$ and the maximum value as $x_{max}$. SmoothGrad introduces a new variable $\alpha$ (set to 0.2 as recommended by [40]) to compute $\sigma$. The relationship between them is expressed in Equation 6.

$$\sigma = \alpha \times (x_{max} - x_{min}) \tag{6}$$

However, we believe that this setup has led to a significant amount of out-of-range sampling behavior during the sampling process, thereby generating extra noise. In fact, SmoothGrad overlooks the fact that the integral in Equation 4 which is not performed over $R^D$, but rather over a bounded domain $\Omega = [x_{min}, x_{max}]$, which is determined by the statistical features of the dataset. In Equation 4, the domains of $G(\cdot)$ and $p(\cdot)$ are inconsistent: the former is defined over $\Omega$, while the latter is defined over $\mathbb{R}^D$. Input samples that fall outside the bounds of $\Omega$ are considered meaningless because they do not align with the statistical properties of the dataset. Consequently, during the smoothing process, SmoothGrad samples values that lie outside $\Omega$, and this out-of-bounds sampling behavior introduces a significant amount of extra noise into $G_{sg}$. Figure 3 provides an example illustrating the presence of this extra noise.

Table 1: The Spearman correlation test results between $OBA$ and $OBV$ with **Sparseness** under different hyperparameter settings

| Correlation coefficient (p-value) | Hyperparameter($\alpha$) | | | | |
|---|---|---|---|---|---|
| Variable | 0.1 | **0.2** | 0.3 | 0.4 | 0.5 |
| $OBA$ | -0.0499(0.1147) | **-0.0551(0.0813)** | -0.0610(0.0538) | -0.0562(0.0757) | -0.0574(0.0696) |
| $OBV$ | -0.0515(0.1038) | **-0.0592(0.0615)** | -0.0625(0.0482) | -0.0579(0.0673) | -0.0586(0.0637) |

Further, we quantitatively define the extra noise as the probability of sampling points that fall outside the domain of the dataset in gradient smoothing methods. This allows us to mathematically express the extra noise introduced in SmoothGrad.

Given an input sample $\mathbf{x} = [x_1, x_2, \cdots, x_D] \in \Omega$, for simplicity, we only focus on one variable $x_i$ of $\mathbf{x}$. Clearly, for the one-dimensional case, the smoothed gradient $G_{sg}^i$ of the gradient with respect to the variable $x_i$ can be represented in Equation 7.

$$G_{sg}^i \simeq \int_{x_{min}}^{x_{max}} G(x_i + \varepsilon_i; \mathbf{x} \backslash x_i) p(\varepsilon_i) d\varepsilon_i, \varepsilon_i \sim \mathcal{N}(0, \sigma^2), \ p = PDF(\mathcal{N}(0, \sigma^2)) \tag{7}$$

Notice that $x_i + \varepsilon_i$ is also a random variable and follows the distribution $\mathcal{N}(x_i, \sigma^2)$. Therefore, we quantify extra noise as the probability that $x_i + \sigma$ falls outside the sampling interval $[x_{min}, x_{max}]$. The extra noise on the i-th dimension $A^i$ can be expressed as Equation 8.

$$A^i = 1 - \int_{x_{min} - x_i}^{x_{max} - x_i} p(t) dt \tag{8}$$

By substituting the expression of SmoothGrad (Equation 5) into Equation 8, we can derive the mathematical expression for the extra noise $A_{sg}^i$ in SmoothGrad, as presented in Equation 9, where $\text{erf}(\cdot)$ denotes the Gaussian Error Function.

$$A_{sg}^i = 1 - \frac{1}{2}\text{erf}(\frac{x_{max} - x_i}{\sqrt{2}\alpha(x_{max} - x_{min})}) + \frac{1}{2}\text{erf}(\frac{x_{min} - x_i}{\sqrt{2}\alpha(x_{max} - x_{min})}) \tag{9}$$

To investigate the correlation between extra noise and out-of-bounds behavior, we employed a noise metric, **Sparseness** [10], and conducted hypothesis testing on this relationship. For a detailed explanation of the Sparseness metric, see Section 5.1.

We utilize two metrics to quantify out-of-bounds behavior: the proportion of out-of-bounds pixels in SmoothGrad (denoted as $OBA$, representing the statistical value of $A_{sg}$) and the sum of values for out-of-bounds pixels (denoted as $OBV$). Additionally, we employ **Sparseness** and VGG16 to evaluate the amount of noise, where a higher **Sparseness** value indicates less noise. We conduct experiments on 1000 samples from ILSVRC2012 and vary the $\alpha$ parameter in SmoothGrad. Table 1 presents the Spearman correlation test results between $OBA$ and $OBV$ with **Sparseness** under all hyperparameter settings. Across all settings, the variables $OBA$ and $OBV$ show a negative correlation with **Sparseness**. Although the absolute value of the Spearman correlation coefficient is very low, which is mainly due to the fact that **Sparseness** is not perfectly positively correlated with the amount of noise, we can accept our hypothesis with 90% confidence at the setting of $\alpha = 0.2$. This indicates a relationship between out-of-bounds behavior and the presence of noise in the smoothed gradients, thereby validating our hypothesis regarding extra noise.

## 4 Adapted Sampling to Reduce Noise

The inconsistency between the noise sampling distribution and the domain of the input data leads to the fact that the smoothed gradient still retains a certain amount of extra noise. Therefore, we propose a gradient smoothing method called AdaptGrad, which adaptively adjusts the noise sampling distribution according to the input data to alleviate this problem and significantly improve the performance of the gradient smoothing.

According to the analysis in Section 3.2, our goal is to control the extra noise. Therefore, one of the most direct methods is to set a minimum upper limit on the amount of extra noise. In fact, this goal is

conceptually similar to parameter estimation parameters under a given confidence level. Following this idea, we design a new gradient smoothing method, AdaptGrad, to generate the smoothed gradient $G_{ag}$ with a specified **extra noise level** $c$. The $G_{ag}$ is computed using Equation 10 - Equation 12, where erfinv($\cdot$) represents the Inverse Gaussian Error Function (the inverse function of erf($\cdot$)) and diag denotes the diagonal matrix.

$$G_{ag} = \frac{1}{N} \sum_{i}^{N} G(\mathbf{x} + \epsilon), \ \epsilon \sim \mathcal{N}^D(0, \Sigma_{ag}) \tag{10}$$

$$\Sigma_{ag} = \text{diag}(\sigma_1^2, \sigma_2^2, \cdots, \sigma_D^2) \tag{11}$$

$$\sigma_i = \begin{cases} \frac{\min(|x_i - x_{min}|, |x_i - x_{max}|)}{\sqrt{2}\text{erfinv}(\frac{1+c}{2})} & \text{if } x_i \neq x_{\min} \text{ and } x_i \neq x_{\max} \\ 0 & \text{if } x_i = x_{\min} \text{ or } x_i = x_{\max} \end{cases} \tag{12}$$

To explain the design idea of AdaptGrad, we continue to focus on one variable $x_i$. Our goal is to calculate $\sigma_i$ such that the random variable $x_i + \varepsilon_i$ falls within the sampling interval $[x_{min}, x_{max}]$ at a extra noise level of $c$. This implies that the maximum allowable extra noise is directly limited to $1 - c$. Therefore, we need to solve the variable $\sigma_i$ in Equation 13.

$$1 - \underbrace{\frac{1}{2}\text{erf}(\frac{x_{max} - x_i}{\sqrt{2}\sigma_i}) + \frac{1}{2}\text{erf}(\frac{x_{min} - x_i}{\sqrt{2}\sigma_i})}_{A_{ag}^i} = c \tag{13}$$

However, the $\sigma_i$ in Equation 13 does not have a simple analytical solution, making it impractical to implement the corresponding algorithm directly. To address this issue, we leverage the symmetry of the normal distribution and define the sampling interval as $[-\min(|x_{max} - x_i|, |x_{min} - x_i|), \min(|x_{max} - x_i|, |x_{min} - x_i|)]$. This sampling interval ensures that the half-length is determined by the shortest distance from $x_i$ to the boundaries. Using this approach, we can derive $\sigma_i$ from Equation 12. In the extended case of $D$-dimensions, we construct the covariance matrix $\Sigma_{ag}$, thereby reducing the noise caused by out-of-bounds behavior via sampling from the distribution $\mathcal{N}^D(0, \Sigma_{ag})$.

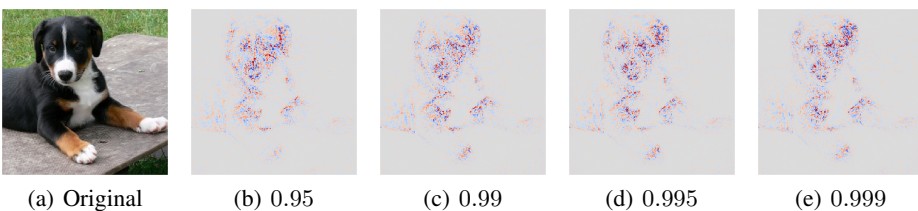

|  |  |  |  |  |
| :---: | :---: | :---: | :---: | :---: |
| (a) Original | (b) 0.95 | (c) 0.99 | (d) 0.995 | (e) 0.999 |

Figure 4: The visual saliency map $G_{ag}$ of AdaptGrad with different extra noise level $c$. Other settings are the same as Figure 2.

In AdaptGrad, the extra noise level $c$ is defined following the concept of low-probability events in probability theory, with typical values such as 0.95, 0.99, 0.995,and 0.999. Figure 4 illustrates the visual results of AdaptGrad at different extra noise levels, highlighting its effectiveness. In Appendix A, we perform a limited hyperparameter search. The results show that AdaptGrad is not only robust to hyperparameter variations but also outperforms SmoothGrad across nearly all hyperparameter configurations. Based on these findings, we recommend using $c = 0.95$ or $c = 0.99$. For consistency, we adopt $c = 0.95$ in all subsequent evaluations. Furthermore, to verify the effectiveness of the AdaptGrad design, which is based on probabilistic inference, we compared its performance with that of a smoothing method that directly clip the sampling according to the sampling interval $[x_{min}, x_{max}]$ in Appendix F.

## 5 Experiments

In this section, we evaluate AdaptGrad and baseline methods from both qualitative and quantitative perspectives, as well as their performance when combined with other explanation methods. To further

evaluate AdaptGrad, we designed indirect experiments, detailed in Appendix E and Appendix F, which provide additional insights into its effectiveness and efficiency.

## 5.1 Experimental Settings

All experimental codes and detailed results can be found in the Supplementary Material, and will be released on the public code platform under the anonymous policy. And more experimental details can be found in Appendix C.

### 5.1.1 Metrics

Gradient smoothing methods are often applied as explanation techniques in the field of computer vision. As such, the quality of visualization is a critical metric for evaluating the effectiveness of these methods. However, there is currently no standardized framework or metric to systematically measure the quality of visualizations. To address this gap, we aim to provide a set of visualization examples in Section 5.2 and Appendix G to objectively demonstrate the effectiveness of AdaptGrad.

Additionally, recent studies [19, 50, 28, 17, 22] proposed a wide range of evaluation metrics that incorporate human evaluation. Among these, we adopt four specific metrics to assess the explanation performance, as they align with the two fundamental objectives of model explanation: understanding model decisions [3, 39, 12, 16, 34] and enhancing human understanding [8, 48, 21, 6, 29]. **Consistency** [4] evaluates whether the explanation method aligns with the model's learning capability. **Invariance** [25] ensures that the explanation method maintains output invariance in the presence of constant data offsets within datasets sharing the same model architecture. **Sparseness** [10] measures the distinguishability and identifiability of the saliency map. **Faithfulness** [19] quantifies the fidelity of salience map to reflect the model's decision-making process. We report the variance of these metrics through 5 independent experiments.

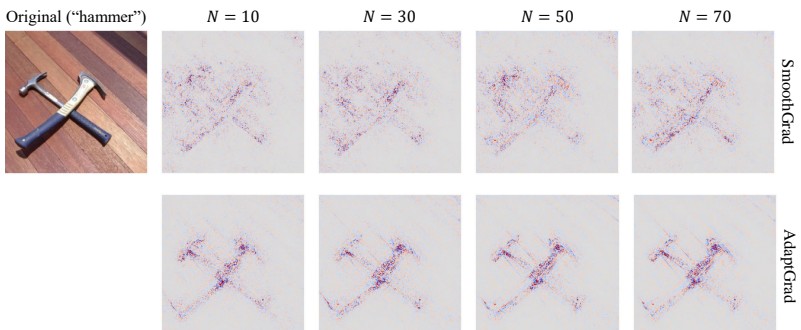

Figure 5: The visual saliency map from VGG16 of SmoothGrad and AdaptGrad with different sample times $N$.

### 5.1.2 Datasets and Models

To apply these metrics for comprehensive evaluation, following the experimental setup in [25, 2], we choose MNIST [27] for experiments on **Consistency** and **Invariance**, ILSVRC2012 (ImageNet) [26] for experiments on **Sparseness** and **Faithfulness**.

Correspondingly, we construct a MLP model for **Consistency** and **Invariance** check, VGG16 [37], ResNet50 [18] and InceptionV3 [43] for visualization, **Sparseness** and **Faithfulness** experiments. VGG16, ResNet50, and InceptionV3 are constructed by pre-trained models released in Torchvision [2]. The MLP architecture consists of two linear layers with 200 and 10 units, respectively. The MLP was trained on MNIST by SGD optimizer with 20 epochs, and the learning rate was set to 0.01.

### 5.1.3 Explanation Methods

AdaptGrad, similar to SmoothGrad, is model-agnostic and can be applied to any gradient-based interpretability methods. Therefore, in addition to using smoothed gradients, we will also use

---

[2]https://pytorch.org/vision/stable/index.html

AdaptGrad alongside other specific methods to generate saliency maps. However, due to the large number of gradient-based explanation methods available, applying AdaptGrad to all of them is computationally challenging. Thus, following [9] and [40], we select three different explanation methods for our experiments.

**Gradient $\times$ Input (GI)** [38], is a simple explanation method that generates saliency maps by directly multiplying the image gradients with the input image. **GI** is the representative of the methods that directly use gradients to generate saliency maps. **Integrated Gradients (IG)** [42] generates saliency maps using global integrated gradients, which can avoid the gradient saturation problem. **IG** is the representative of the methods that partially use gradients in their computation. **IG** has different options for the baseline background. We have chosen black and white as the baseline backgrounds, which are labeled as **IG(B)** and **IG(W)** respectively. **NoiseGrad (NG)** [9] is another gradient smoothing method. Unlike SmoothGrad and AdaptGrad, NG reduces noise by perturbing model parameters. However, this approach incurs significant computational costs and does not substantially improve visualization quality. **NG** represents other gradient smoothing methods. To denote the combination of SmoothGrad and AdaptGrad with other methods, we use the prefixes **S-** and **A-**, respectively. **Grad**, **SG**, and **AG** denote the original gradient, SmoothGrad, and AdaptGrad.

Referring to the setup in [2, 25, 9], **Consistency** and **Invariance** are employed to validate SmoothGrad, AdaptGrad, and their combinations with **NG**. While **Sparseness** is applied to evaluate all the explanation methods. Since **Faithfulness** is divided into two types of scores: insertion scores and deletion scores, we use **Faithfulness-I** and **Faithfulness-D** to label these two types of scores respectively.

## 5.2  Qualitative Evaluation

As shown in Figure 5, we compare the visualization effects of SmoothGrad and AdaptGrad using different sampling numbers $N$. The saliency maps generated by AdaptGrad demonstrate better visualization quality than those produced by SmoothGrad, particularly in terms of the clarity and detail of object representations. Even at a low sampling number ($N = 10$), AdaptGrad can exhibit a clear noise reduction capability.

In Figure 6, we present an example of saliency maps generated by applying SmoothGrad and AdaptGrad to the methods **GI**, **IG(B)**, **IG(W)**, and **NG**. The results demonstrate that AdaptGrad has a clear advantage over SmoothGrad in visualizing latent features. Specifically, AdaptGrad provides a more nuanced and detailed representation. Furthermore, the enhancement effect of AdaptGrad on the **IG** method is pronounced, with a notable reduction in noise in the saliency map and the presentation of intricate detail features, such as the facial features of the object.

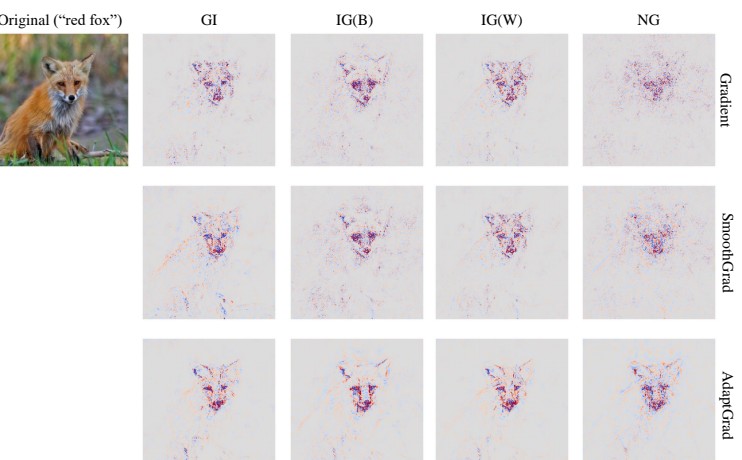

Figure 6: The visual saliency map from VGG16 of Gradient, SmoothGrad, and AdaptGrad combined with **GI**, **IG(B)**, **IG(W)** and **NG**.

Table 2: Results of **Consistency** and **Invariance** checks for SmoothGrad (**SG**) and AdaptGrad (**AG**)

| Methods | Grad | SG | AG |
|---|---|---|---|
| Consistency | 0.02076(0.00028) | 0.01911(0.00014) | 0.020239(0.00026) |
| Invariance | 0.3483(0.0002) | 0.3613(0.0009) | 0.3484(0.0002) |

Table 3: Results of **Sparseness** (SS), **Faithfulness-I** (FI) and **Faithfulness-D** (FD) evaluation for VGG16. The ↑ indicates the higher is better.

| Metrics | Value | Grad | SG | AG | GI | S-GI | A-GI | IG(W) | S-IG(W) | A-IG(W) | IG(B) | S-IG(B) | A-IG(B) | NG | S-NG | A-NG |
|---|---|---|---|---|---|---|---|---|---|---|---|---|---|---|---|---|
| SS(↑) | Mean | 0.5583 | 0.5289 | **0.5740** | 0.6417 | 0.6137 | **0.6821** | 0.5535 | 0.5814 | **0.5901** | 0.5765 | 0.6015 | **0.6168** | 0.5669 | 0.5942 | **0.6203** |
| | (Var.) | (0.0000) | (0.0001) | (0.0000) | (0.0000) | (0.0000) | (0.0001) | (0.0000) | (0.0000) | (0.0000) | (0.0000) | (0.0000) | (0.0000) | (0.0004) | (0.0003) | (0.0009) |
| FI(↑) | Mean | **0.6830** | 0.6729 | 0.6748 | **0.6672** | 0.5629 | 0.5782 | 0.6503 | 0.6471 | **0.6585** | 0.6549 | 0.6447 | **0.6656** | **0.6872** | 0.6654 | 0.6724 |
| | (Var.) | (0.0000) | (0.0003) | (0.0002) | (0.0000) | (0.0005) | (0.0012) | (0.0000) | (0.0004) | (0.0000) | (0.0000) | (0.0004) | (0.0003) | (0.0001) | (0.0004) | (0.0003) |
| FD(↓) | Mean | 0.6830 | **0.6728** | 0.6747 | 0.6672 | **0.5628** | 0.5781 | 0.6502 | **0.6406** | 0.6584 | 0.6548 | **0.6447** | 0.6656 | 0.6873 | **0.6653** | 0.6724 |
| | (Var.) | (0.0000) | (0.0003) | (0.0002) | (0.0000) | (0.0005) | (0.0011) | (0.0000) | (0.0004) | (0.0001) | (0.0000) | (0.0004) | (0.0003) | (0.0001) | (0.0004) | (0.0003) |

Table 4: Results of **Sparseness** (SS), **Faithfulness-I** (FI) and **Faithfulness-D** (FD) evaluation for InceptionV3. The ↑ indicates the higher is better.

| Metrics | Value | Grad | SG | AG | GI | S-GI | A-GI | IG(W) | S-IG(W) | A-IG(W) | IG(B) | S-IG(B) | A-IG(B) | NG | S-NG | A-NG |
|---|---|---|---|---|---|---|---|---|---|---|---|---|---|---|---|---|
| SS(↑) | Mean | 0.5441 | 0.5369 | **0.5584** | 0.6215 | 0.6108 | **0.6547** | 0.5595 | 0.5666 | **0.5751** | 0.5778 | 0.5867 | **0.6043** | 0.4661 | 0.4538 | **0.4669** |
| | (Var.) | (0.0000) | (0.0001) | (0.0001) | (0.0000) | (0.0000) | (0.0001) | (0.0000) | (0.0000) | (0.0000) | (0.0000) | (0.0000) | (0.0000) | (0.0017) | (0.0016) | (0.0016) |
| FI(↑) | Mean | **0.6246** | 0.5955 | 0.6145 | **0.6249** | 0.4317 | 0.5098 | 0.5860 | 0.5920 | **0.6138** | 0.5837 | 0.5906 | **0.6257** | **0.5696** | 0.5504 | 0.5676 |
| | (Var.) | (0.0002) | (0.0016) | (0.0007) | (0.0004) | (0.0032) | (0.0024) | (0.0001) | (0.0006) | (0.0006) | (0.0001) | (0.0004) | (0.0003) | (0.0037) | (0.0032) | (0.0040) |
| FD(↓) | Mean | 0.6247 | **0.5955** | 0.6146 | 0.6247 | **0.4318** | 0.5099 | **0.5860** | 0.5920 | 0.6138 | **0.5837** | 0.5906 | 0.6257 | 0.5696 | **0.5504** | 0.5676 |
| | (Var.) | (0.0002) | (0.0015) | (0.0008) | (0.0004) | (0.0031) | (0.0025) | (0.0001) | (0.0006) | (0.0006) | (0.0001) | (0.0004) | (0.0003) | (0.0037) | (0.0033) | (0.0039) |

Table 5: Results of **Sparseness** (SS), **Faithfulness-I** (FI) and **Faithfulness-D** (FD) evaluation for ResNet50. The ↑ indicates the higher is better.

| Metrics | Value | Grad | SG | AG | GI | S-GI | A-GI | IG(W) | S-IG(W) | A-IG(W) | IG(B) | S-IG(B) | A-IG(B) | NG | S-NG | A-NG |
|---|---|---|---|---|---|---|---|---|---|---|---|---|---|---|---|---|
| SS(↑) | Mean | 0.5536 | 0.5614 | **0.5721** | 0.6370 | 0.6320 | **0.6703** | 0.5536 | 0.5902 | **0.6003** | 0.5710 | 0.6051 | **0.6115** | 0.4785 | 0.4695 | **0.4912** |
| | (Var.) | (0.0000) | (0.0001) | (0.0001) | (0.0000) | (0.0001) | (0.0001) | (0.0000) | (0.0000) | (0.0001) | (0.0000) | (0.0000) | (0.0001) | (0.0057) | (0.0107) | (0.0088) |
| FI(↑) | Mean | **0.2767** | 0.2626 | 0.2692 | **0.2757** | 0.1002 | 0.1496 | 0.2590 | 0.2747 | **0.2940** | 0.2665 | 0.2703 | **0.2954** | 0.2618 | 0.2472 | 0.2479 |
| | (Var.) | (0.0001) | (0.0011) | (0.0011) | (0.0003) | (0.0019) | (0.0002) | (0.0001) | (0.0005) | (0.0004) | (0.0000) | (0.0003) | (0.0002) | (0.0013) | (0.0018) | (0.0014) |
| FD(↓) | Mean | 0.2767 | **0.2626** | 0.2693 | 0.2756 | **0.1002** | 0.1496 | **0.2590** | 0.2747 | 0.2940 | **0.2665** | 0.2703 | 0.2954 | 0.2618 | **0.2472** | 0.2479 |
| | (Var.) | (0.0001) | (0.0011) | (0.0011) | (0.0003) | (0.0019) | (0.0002) | (0.0001) | (0.0005) | (0.0004) | (0.0000) | (0.0003) | (0.0002) | (0.0013) | (0.0018) | (0.0014) |

## 5.3 Quantitative Evaluation

The settings outlined in Section 5.1 are employed to initially assess the **Consistency** and **Invariance** of AdaptGrad. The metric values are evaluated to determine whether they exhibit any unusual deviations compared to the original gradients. The results, as shown in Table 2, indicate that both AdaptGrad and SmoothGrad fall within the normal range for **Consistency** and **Invariance**. This suggests that AdaptGrad successfully meets the criteria for these two metrics.

Table 3, Table 4 and Table 5 reveals that AdaptGrad demonstrates significant improvement in the evaluation of **Sparseness** and **Faithfulness**. In terms of Sparseness, AdaptGrad shows a clear advantage over the SmoothGrad method across all 3 models and 5 types of interpretability methods. For the **Faithfulness**, AdaptGrad also mostly outperforms SmoothGrad. This indicates that AdaptGrad is capable to achieve a better balance between the visual quality of the significance maps and the fidelity of the model.

However, as noted by [49], the **Faithfulness** metric is highly dependent on the model's inherent performance and suffers from a pronounced long-tail effect. In addition, this metric involves numerous hyperparameter choices, which further make its evaluation process less fair and consistent. Therefore, we argue that Faithfulness may not serve as an accurate indicator of an explanation method's true quality. The details and limitations of this metric are further discussed in Appendix B.

These metrics rely on certain assumptions about the quantitative analysis of visualization effects, which means they can only indirectly evaluate the performance of interpretability methods. To provide more direct evidence, Appendix G includes numerous illustrative visualizations that highlight AdaptGrad's superior denoising capability.

## 6 Conclusion

In this paper, we reconsider the principles of gradient smoothing methods by applying the convolution formula, which helps identify the presence of extra noise in gradient smoothing. Based on this analysis, we propose an adaptive gradient smoothing method, AdaptGrad, designed to mitigate extra noise. Theoretical analyses of extra noise, supported by qualitative and quantitative experiments,

demonstrate that AdaptGrad is an effective alternative to SmoothGrad. Specifically, AdaptGrad outperforms SmoothGrad in terms of noise reduction and robustness. In terms of implementation, AdaptGrad, like SmoothGrad, is computationally efficient and model-agnostic. It can also be integrated with other gradient-based explanation methods to enhance their performance.

**Limitations and Future works**

*Selection of hyperparameters.* In fact, the extra noise level $c$ in AdaptGrad is still an empirical choice, despite its widespread use in the field of probability theory. For different datasets or artificial intelligence tasks, there should be different optimal parameter choices.

*Measurement of noise.* In Section 3.2, we used the **Sparseness** metric to evaluate the amount of noise present in the saliency map. Since there are no established methods for directly assessing noise, we relied on a single indirect evaluation approach. Moreover, we argue that the **Faithfulness** metric based on insertion and deletion scores is not a reliable measure of a saliency map's explanatory capability. This limitation may result in insufficient experimental evidence to empirically demonstrate the relationship between out-of-bounds sampling behavior and noise.

*Evaluation of explanation methods.* A comprehensive and fair evaluation of explanation methods is a common challenge in current research on explanation methods. As a result, we also face this issue in our work. To advance research in this field, we hope to develop a unified and widely accepted evaluation framework to provide clear guidelines for assessment.

*Broader impact of AdaptGrad.* Through comprehensive experimental design, we demonstrate that AdaptGrad provides a more efficient yet simple approach to gradient denoising. Our comparative experiments show that integrating AdaptGrad with existing interpretation methods such as Integrated Gradients and NoiseGrad can significantly improve their performance. This suggests that AdaptGrad can enhance the explanatory power of nearly all gradient-based interpretation methods. Beyond the three gradient-dependent methods validated in our experiments, other approaches such as Smooth-CAM++[33], Smooth Score-CAM[45], IDGI[49], and TAIG[20] can also directly replace their gradient smoothing modules with AdaptGrad to achieve better denoising performance. Therefore, we hope that by actively contributing to the open-source community, we can provide new solutions to advance the development of the XAI field.

# Acknowledgments and Disclosure of Funding

This work is supported by National Natural Science Foundation of China (No.62272352, 63441237, U24A20336). The authors thank for the helpful discussions of anonymous reviewers and area chairs.

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

# A  Hyperparameters Selection

As presented in Equation 10-Equation 12, AdaptGrad only contains two hyperparameters, sample times $N$ and extra noise level $c$. And SmoothGrad also only contains two hyperparameters, sample times $N$ and $\alpha$. All examples and experiments in this article use the settings of $N = 50, c = 0.95, \alpha = 0.2$. The settings for SmoothGrad refer to [40], while the settings for AdaptGrad simply follow conventions in probability theory. We did not employ any hyperparameter optimization or search methods in the experiments.

Clearly, the setting of hyperparameters can affect the performance of AdaptGrad and SmoothGrad. However, the excellent performance of AdaptGrad is robust to the selection of hyperparameters. To prove this, we use **Sparseness**, as mentioned in Section 5.2 (which is related to visualization performance), as an indicator to measure the performance for different hyperparameter configurations. Table 6 shows the experimental results, and all experimental configurations are consistent with those in Section 5.2. From the results in Table 6, under the same number of sampling times, the performance of AdaptGrad outperforms that of SmoothGrad with any setting of $c$.

So, we intuitively applied $c = 0.95$ as the setting for all examples and experiments in this paper. Although we do not believe this is the best hyperparameter selection, we believe that regardless of the chosen value of $c$, as long as it falls within a reasonable range, like $0.9 - 0.999$, AdaptGrad is likely to achieve convincing performance.

Another set of hyperparameters is $X_{min}$ and $X_{max}$. $X_{min}$ and $X_{max}$ are generally independent of the dataset. In the field of image processing, it is almost conventional to set the value corresponding to black pixels as $X_{min}$ and the value corresponding to white pixels as $X_{max}$. Therefore, whether it's the SmoothGrad method (seen in Equation 6) or the Integrated Gradients method (seen in Section 5.1.3), this is used as a default assumption in these methods. Therefore, in AdaptGrad, we assume by default that $X_{min}$ and $X_{max}$ are universal information, requiring neither special computational procedures nor manual configuration.

Table 6: The performance (**Sparseness** ↑) of AdaptGrad (AG) and SmoothGrad (SG) with different hyperparameter combinations. We marked the maximum value of SG performance, the minimum value of AG performance, and the maximum value of AG performance using red, blue, and green, respectively, under the same sampling number ($n$).

| SG | **Sparseness** Score (↑) tested on VGG16 | | | | | |
|---|---|---|---|---|---|---|
| $n$ \ $\alpha$ | 10 | 20 | 30 | 50 | 70 | 100 |
| 0.1 | 0.5304 | 0.5330 | 0.5363 | 0.5427 | 0.5475 | 0.5533 |
| 0.2 | 0.5370 | 0.5345 | 0.534 | 0.5334 | 0.5338 | 0.5343 |
| 0.3 | 0.5361 | 0.5313 | 0.5279 | 0.5235 | 0.5209 | 0.5174 |
| 0.4 | 0.5334 | 0.5266 | 0.5221 | 0.5154 | 0.5109 | 0.5057 |
| 0.5 | 0.5309 | 0.5237 | 0.5185 | 0.5113 | 0.5061 | 0.5001 |
| AG | **Sparseness** Score (↑) tested on VGG16 | | | | | |
| $n$ \ $c$ | 10 | 20 | 30 | 50 | 70 | 100 |
| 0.9 | 0.5496 | 0.5493 | 0.5511 | 0.5535 | 0.5561 | 0.5588 |
| 0.95 | 0.5511 | 0.5532 | 0.5558 | 0.5608 | 0.5644 | 0.5687 |
| 0.99 | 0.5526 | 0.5576 | 0.5621 | 0.5691 | 0.5748 | 0.5809 |
| 0.995 | 0.5527 | 0.5582 | 0.563 | 0.5711 | 0.5772 | 0.5838 |
| 0.999 | 0.5526 | 0.5597 | 0.5655 | 0.5746 | 0.5807 | 0.5881 |

# B  Metrics Details

Currently, there is no unified and widely accepted system for the quantitative evaluation of explanation methods. In fact, nearly every related study employs different evaluation metrics, making it difficult to follow a consistent standard for selecting assessment metrics. We conducted a relatively comprehensive evaluation from two perspectives: the axiomatic properties of explanation methods (understanding model decisions) and their visualization effects (enhancing human understanding). Below, we provide a detailed introduction to the origins and implementations of the four quantitative metrics used in this paper. And all the implementations of the metrics are included in our source code.

**Consistency** is from the Sanity Check experiment in [2]. Two types of check experiment, model parameter randomization test and data randomization test, were designed to evaluation Gradient, SmoothGrad, Gradient × Input, Guided Back-propagation, GradCAM, Guided GradCAM, Integrated Gradients, and Integrated Gradients-SG. The model parameter randomization test primarily check whether the explanation method can remain

Table 7: Results of SIC evaluation for SmoothGrad((SG)) and AdaptGrad(AG) combined with (IG). The ↑ indicates the higher is better.

| Methods | VGG16 | InceptionV3 | ResNet50 |
|---------|-------|-------------|----------|
| IG(W) | 0.5636 | 0.6677 | 0.5491 |
| S-IG(W) | 0.5741 | 0.7179 | 0.5718 |
| A-IG(W) | **0.5846** | **0.7221** | **0.5849** |
| IG(B) | 0.5823 | 0.6751 | 0.5355 |
| S-IG(B) | 0.6035 | 0.7166 | **0.5731** |
| A-IG(B) | **0.6121** | **0.7193** | 0.5673 |

consistent with the model's randomization process through visualization effects. While the data randomization test evaluates the consistency, quantified using rank correlation, of the interpretation method before and after randomization by permuting the training labels and training a model on the randomized training data. Therefore, we selected the data randomization test as the evaluation metric in this paper and, following [19], named it **Consistency**.

**Invariance** is from the the axiom input invariance in [25]. They designed a experiment aimed at validate the input shift invariance of explanation methods. Since [25] did not provide a clear name for it, and both [9] and [19] refer to this metric as Robustness. To distinguish it from the adversarial example generation experiments in Appendix E, we named it **Invariance**.

**Sparseness** is from the Assumption LOSS-CVX and sparseness of an attribution vector experiment in [10]. For an explanation method, Gini Index is used to quantify the sparseness of absolute values of saliency maps. Given an input of saliency map $\mathbf{x} = \{x_1, x_2, ..., x_n\}$, its Gini Index $G$ can be calculated using Equation 14 .

$$G = \frac{\sum_{i=1}^{n}(2i - n - 1)x_i}{n\sum_{i=1}^{n} x_i} \tag{14}$$

**Faithfulness** is evaluated using the Area Under the Curve (AUC) of the insertion and deletion metrics, which are widely adopted for assessing the reliability of model explanations. In our experiments, we set the insertion and deletion ratio to 5% of the total pixels, and used 0 as the baseline filling value. Although prior work [49] has pointed out that such straightforward insertion–deletion procedures may not fully capture the faithfulness of an explanation method, we employ them here for fair comparison with existing studies. Interestingly, when pixels are removed in batches according to their saliency ranking, noisier saliency maps may achieve deceptively higher deletion scores. This is because, after removing pixels with high importance, the residual noise tends to "smooth out" the deletion process by also eliminating pixels surrounding truly important regions. This phenomenon aligns with our empirical observations reported in Table 3, Table 4, and Table 5.

Therefore, for Integrated Gradients-based explanation methods, we additionally adopt an improved variant of the insertion–deletion metric, namely the Softmax Information Curve (SIC) proposed by [23]. Unlike conventional insertion and deletion metrics that directly replace pixels with zeros, SIC progressively removes or restores pixels identified as important based on the amount of information they contribute to the model's prediction. We report the experimental results in Table 7, which show that AdaptGrad consistently outperforms other explanation methods under the SIC metric.

## C    Experimental Details

In the quantitative evaluation, due to the time-consuming computation of the experiment, we randomly sampled 1,000 samples instead of all validation or test set samples for the comparison experiments. This also led to difficulties in reporting the statistical significance of our experimental results. Our experiments were conducted on a server with 4×NVIDIA RTX 4090 and 2×Intel Xeon Gold 6128. Although AdaptGrad consumes little computational resources, combining it with other rendering methods results in exponential consumption (similar to SmoothGrad), so we recommend using GPU devices with at least 12G memory to run the reproducible code.

The MLP architecture in **Consistency** and **Invariance** check consists of two linear layers with 200 and 10 units, respectively. The MLP was trained on MNIST by the SGD optimizer with 20 epochs, and the learning rate was set to 0.01. The CNN architecture contains a few layers as follows: [Conv(6), Maxpool(2), Conv(16), Maxpool(2), Linear(120), Linear(84), Linear(10)].

We set the hyperparameters of the **IG** and **NG** explanation methods as described in [42] and [9]. The number of Riemann integration samples in the **IG** was set to 50, and the number of parameter perturbations in the NG was set to 50. The variance of the corresponding perturbation noise was set to 0.2, and numerical overflow perturbations were excluded.

To ensure the stability of the results of metrics **Faithfulness**, each sample was tested 5 times. The values reported in Table 3, Table 4 and Table 5 are the average of 1000 samples tested 5 times. And its hyperparameter, the saliency threshold, was set to [0.05, 0.1, 0.15, 0.2, 0.25, 0.3] according to [23].

# D   Discussion of Hard Threshold

As discussed in Section 3.2, we identified the presence of extra noise in SmoothGrad. Correspondingly, one might consider directly applying a hard constraint to the input samples as a potential way to suppress this extra noise. However, we argue that such a straight hard-threshold operation can hinder the convergence of the smoothing convolution, thereby introducing additional artifacts and noise into the saliency maps.

In contrast, AdaptGrad maintains a convergent sampling process and adaptively adjusts the sampling range, allowing it to better control the extra noise and emphasize the visualization performance. To validate this hypothesis, we implemented a simple variant of SmoothGrad that clips the sampled values within the range $[x_{min}, x_{max}]$, which we refer to as ClipGrad. We then compared ClipGrad and AdaptGrad across all evaluation metrics.

As shown in Table 8 and Table 9, the results consistently demonstrate that AdaptGrad achieves superior performance in both noise suppression and target feature enhancement, confirming its effectiveness over the simple clipping-based alternative.

Table 8: Experimental result of **Consistency** and **Invariance** checks of AdaptGrad (**AG**) and ClipGrad (**CG**).

| Methods | AG | CG |
|---|---|---|
| Consistency | 0.20239(0.00026) | 0.01773(0.00013) |
| Invariance | 0.3484(0.0002) | 0.3625(0.0008) |

Table 9: Results of **Sparseness** (SS), **Faithfulness-I** (FI) and **Faithfulness-D** (FD) evaluation for AdaptGrad (**AG**) and ClipGrad (**CG**). The ↑ indicates the higher is better.

| Models | VGG16 | | InceptionV3 | | ResNet50 | |
|---|---|---|---|---|---|---|
| Metrics | AG | CG | AG | CG | AG | CG |
| SS(↑) | **0.5740(0.0000)** | 0.5294(0.0001) | **0.5584(0.0001)** | 0.5441(0.0001) | **0.5721(0.0001)** | 0.5607(0.0001) |
| FI(↑) | **0.6748(0.0002)** | 0.6740(0.0006) | **0.6145(0.0007)** | 0.5999(0.0005) | **0.2692(0.0011)** | 0.2605(0.0020) |
| FD(↓) | 0.6747(0.0002) | **0.6739(0.0006)** | 0.6146(0.0008) | **0.5999(0.0006)** | 0.2693(0.0011) | **0.2605(0.0019)** |

# E   Evaluation on Visual Task

In this section, we compare the differences between AdaptGrad and SmoothGrad in two common visual tasks that utilize saliency maps: object localization and adversarial sample generation. These indirect visual tasks could assist in evaluating the performance of the AdaptGrad method more objectively.

## E.1   Evaluation on Object Localization Task

The performance improvement in weakly supervised object localization tasks is often used to evaluate class activation map methods as a type of explanation method. [54, 35, 11, 46, 22] generate saliency maps based on explanation methods and use "heuristic" approaches to create a series of object localization candidate bounding boxes. Then the localization accuracy of these candidate bounding boxes is used to evaluate explanation methods. Due to the numerous uncertainties associated with these "heuristic" methods, we employ a publicly available and widely used candidate bounding box generation algorithm called selective search[44]. The selective search algorithm has a very high recall rate but a lower precision rate. Therefore, we used its precision rate to measure the performance of AdaptGrad and SmoothGrad.[3]

The hyperparameter settings of this experiment are consistent with other experiments. Specifically, the parameters of the selective search algorithm are set as follows: clusters number in felzenszwalb segmentation is 500 (scale=500), width of Gaussian kernel is 0.9 (sigma=0.9), and minimum component size is 10 (min_size=10). Figure 7 illustrates an example of candidate bounding boxes using the selective search algorithm. It is clearly observed that the candidate bounding boxes generated based on AdaptGrad are more concentrated on the ground truth box.

---

[3]The saliency map generated by the vanilla gradients contains a huge amount of noise, which causes the selective search algorithm unable to generate usable bounding candidate boxes. Most of these bounding boxes only contains a few pixels. So we exclude the testing for Vanilla Gradient.

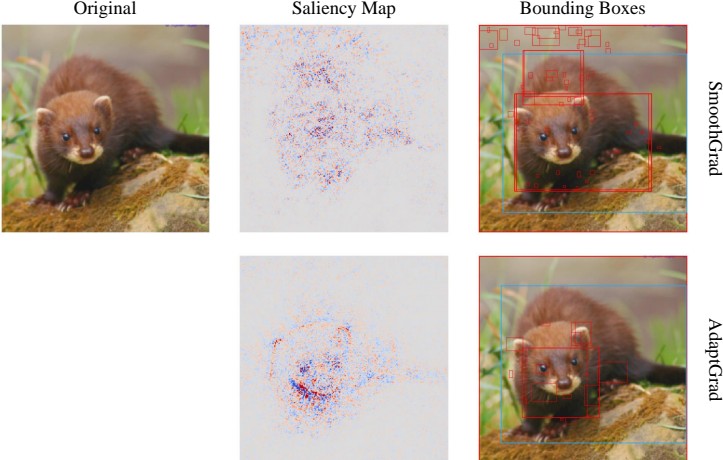

Figure 7: The visual bounding boxes from VGG16 of SmoothGrad and AdaptGrad with selective search algorithm. The generated candidate bounding boxes are marked in red, and the ground truth box is marked in blue.

Table 10: The object localization bounding boxes precision, which is generated by selective search algorithm using saliency maps from SmoothGrad and AdaptGrad.

| Models | Localization Precision (↑) | | |
|---|---|---|---|
| | VGG16 | InceptionV3 | ResNet50 |
| SmoothGrad | 0.01937 | 0.1048 | 0.05528 |
| AdaptGrad | **0.05951** | **0.1237** | **0.08108** |

Table 10 summarizes the performance of SmoothGrad and AdaptGrad in our designed object localization capability evaluation. It can be observed that AdaptGrad outperforms SmoothGrad across all three test models. This indicates that AdaptGrad could more accurately represent the neural network's learning capability for object features, and also suggests that AdaptGrad may have better potential applications in weakly supervised object localization algorithms.

## E.2 Evaluation on Adversarial Sample Generation Task

In this section, we indirectly evaluate the performance of different explanation methods by comparing their performance in the adversarial sample generation task. We use the pixel-level Fast Gradient Sign Method (FGSM) to generate adversarial samples. Specifically, we select the pixel corresponding to the maximum value in the saliency map (i.e., the gradient map in FGSM) of a sample one by one and change the value of this pixel to make the neural network incorrectly classify the sample.

We adopt pixel-level FGSM because if we generate adversarial samples directly based on the entire gradient map, it would lead to noisier gradients performing better (i.e., causing a greater drop in model accuracy), as this noise causes more pixels to change in the adversarial sample. However, this is clearly not the sole objective of the adversarial sample generation task. The quality evaluation of adversarial samples should also consider the similarity between the adversarial sample and the original image, as well as the subjective image quality. Therefore, to avoid these complex adversarial sample quality evaluation issues, we generate adversarial samples based on pixel-level FGSM. By setting a uniform attack target, we then measure the performance of different explanation methods by counting the number of pixels that need to be changed based on the saliency map generated by each explanation method. In our experiment, we set the attack target to make the model's output for the target class (Softmax output) less than 0.5.

Figure 8 shows an example of an adversarial sample. Due to the high randomness of adversarial attacks, we chose a relatively simple VGG16 model as the attack target. We comprehensively evaluate the performance of different explanation methods in the adversarial sample generation task by conducting 10 independent experiments. The rest of the experimental settings remain consistent with other experiments in this paper. Figure 9 illustrates the experimental results based on the evaluation of the adversarial sample generation task. The FGSM method based on the original gradient exhibited significant volatility. However, it can be observed that the FGSM method based on AdaptGrad requires significantly fewer pixel changes compared to the FGSM method based on SmoothGrad. This indicates that AdaptGrad can more accurately reveal the internal decision mechanisms of the

model compared to SmoothGrad, and suggests that AdaptGrad may have potential applications in adversarial sample generation.

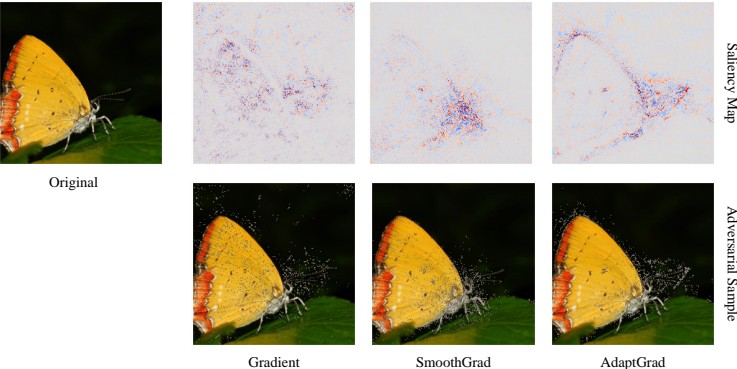

Figure 8: Comparison of adversarial samples generated by different explanation methods. The attacked model is VGG16.

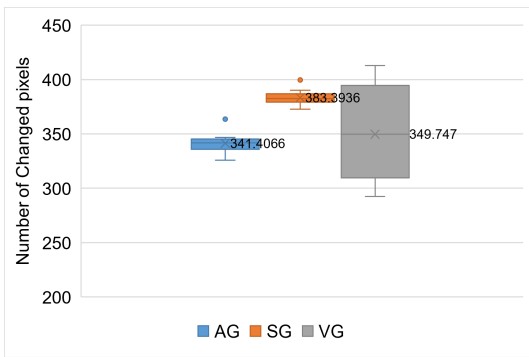

Figure 9: Statistics on the number of pixels that need to be changed to generate adversarial samples based on different interpretation methods. AG, SG and VG represent AdaptGrad, SmoothGrad and Vanilla Gradient respectively.

# F  Evaluation on Computational Cost

Compared to SmoothGrad, AdaptGrad introduces only a minimal amount of additional computation. This extra computation is almost entirely concentrated in Equation 12. However, it's worth noting that in Equation 12, the denominator can be practically treated as a constant, while the computational complexity of the numerator only depends on the dimension of the input, with a complexity of just $O(N)$. Therefore, AdaptGrad only adds an $O(N)$ level of complexity, which is virtually negligible compared to the computation of the gradients of the neural network itself.

To demonstrate this, we tested the computational time of SmoothGrad and AdaptGrad on benchmark models (VGG16, InceptionV3, ResNet50) using 1,000 images. The hardware and other parameter settings were entirely consistent with the other experiments. The Table 11 below presents our experimental results.

Table 11: Execution time (s) of AdaptGrad and baselines.

| Method | VGG16 | InceptionV3 | ResNet50 |
|---|---|---|---|
| Grad | $0.0075 \pm 0.0044$ | $0.0387 \pm 0.0092$ | $0.01828 \pm 0.0069$ |
| SmoothGrad | $0.5914 \pm 0.0162$ | $2.6255 \pm 0.0724$ | $1.9955 \pm 0.0877$ |
| AdaptGrad | $0.6054 \pm 0.0201$ | $2.6673 \pm 0.0829$ | $1.9426 \pm 0.0873$ |

The results show that AdaptGrad incurs only a very small extra computational overhead. This demonstrates the strong versatility of AdaptGrad and SmoothGrad. We will also include the corresponding experimental results and analysis in the paper, and provide a detailed discussion on the potential computational costs when dealing with large-scale data and complex models.

# G More Visualization Examples

We provide more visualization examples to compare AdaptGrad with the Baseline method, including experiments on the VGG16, ResNet50, and InceptionV3 models. And we provide visualization examples on images with low contrast, high contrast, small targets, large targets, and multiple targets

Figure 10, Figure 11 and Figure 12 show the explanation performance of AdaptGrad and baselines on VGG16, ResNet50, and InceptionV3 models. It can be noticed that AdaptGrad could improve the visualization performance.

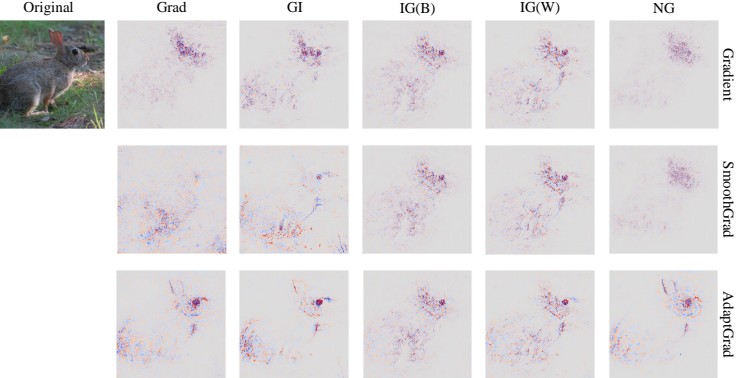

Figure 10: The visual saliency map from VGG16 of Gradient, SmoothGrad, and AdaptGrad combined with Grad, GI, IG(B), IG(W) and NG.

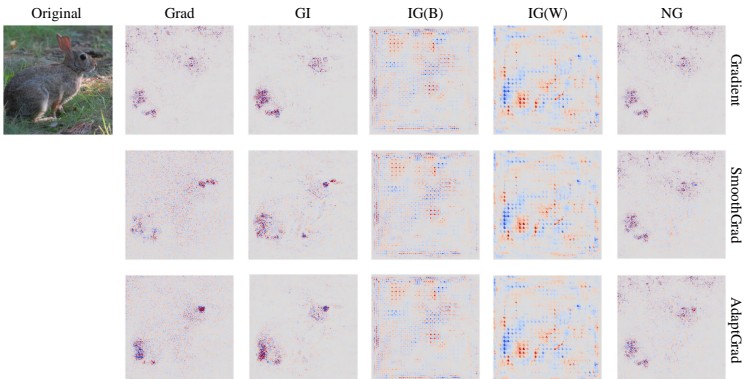

Figure 11: The visual saliency map from ResNet50 of Gradient, SmoothGrad, and AdaptGrad combined with Grad, GI, IG(B), IG(W) and NG.

We also conducted visual demonstrations on various types of images, including high-contrast images Figure 13, low-contrast images Figure 14, large-object images Figure 15, small-object images Figure 16, and multi-object images Figure 17 using VGG16. All these images were sourced from the ImageNet dataset. Specifically, the high-contrast images were selected from among the highest-contrast images in the ImageNet validation set, and the same applies to the low-contrast images, large-object images, and small-object images. Almost all the examples demonstrate that AdaptGrad can achieve better visualization results than the baseline on different types of images.

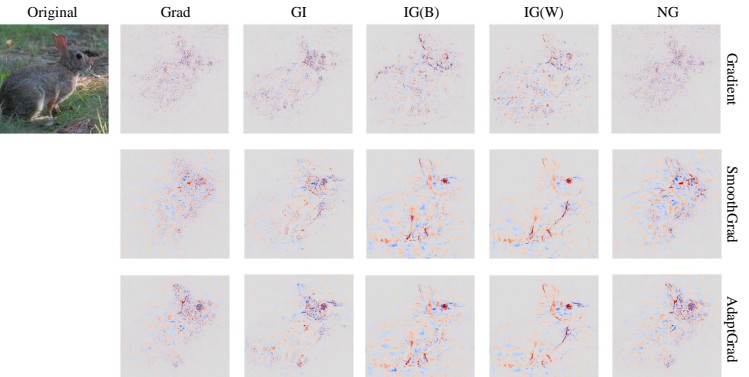

Figure 12: The visual saliency map from InceptionV3 of Gradient, SmoothGrad, and AdaptGrad combined with Grad, GI, IG(B), IG(W) and NG.

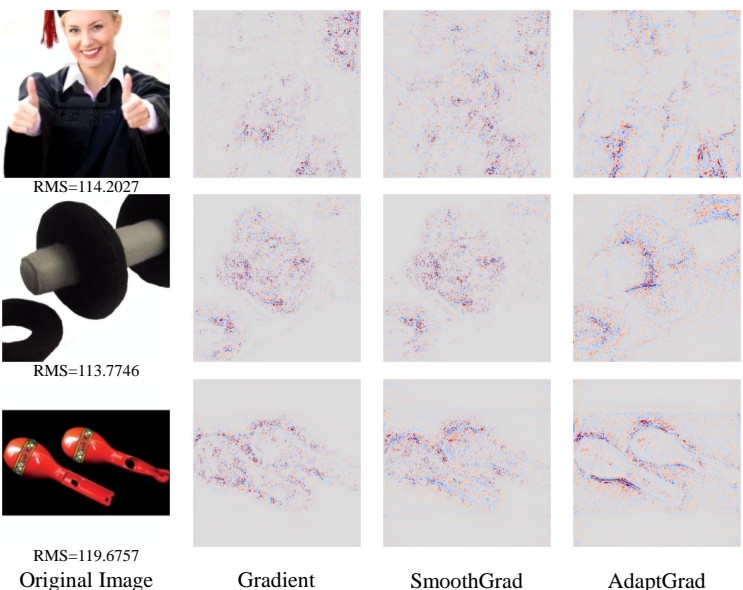

Figure 13: The visual saliency map of high-contrast images comparison between Gradient, Smooth-Grad, and AdaptGrad

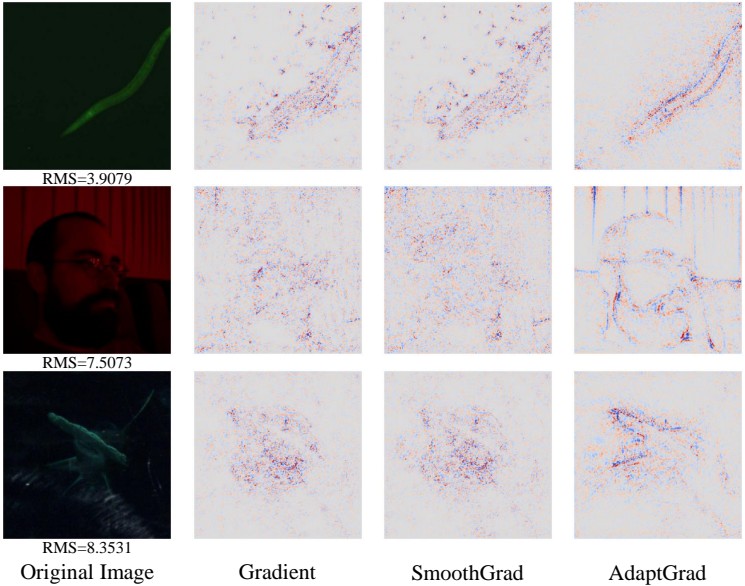

Figure 14: The visual saliency map of low-contrast images comparison between Gradient, Smooth-Grad, and AdaptGrad

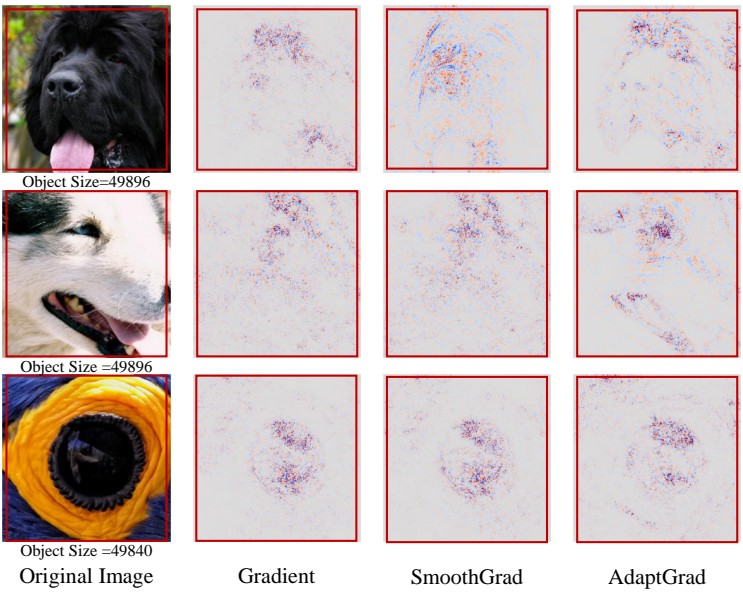

Figure 15: The visual saliency map of large-object images comparison between Gradient, Smooth-Grad, and AdaptGrad

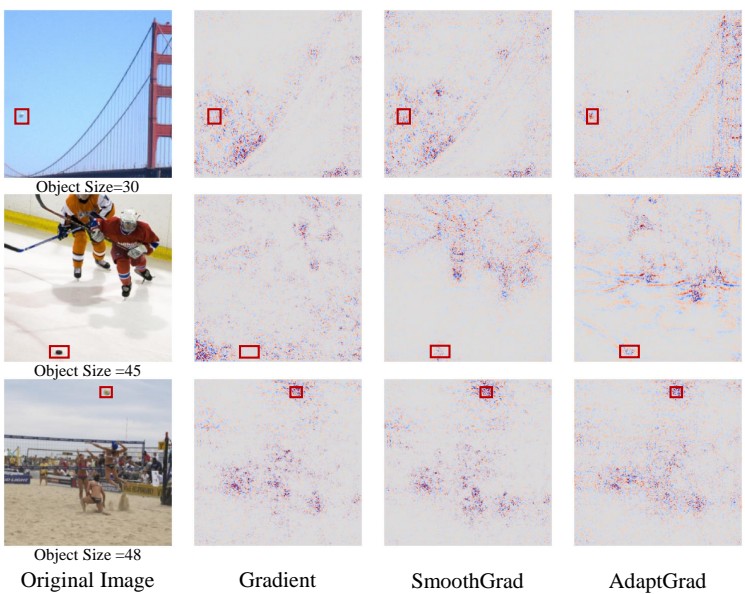

Figure 16: The visual saliency map of small-object images comparison between Gradient, Smooth-Grad, and AdaptGrad

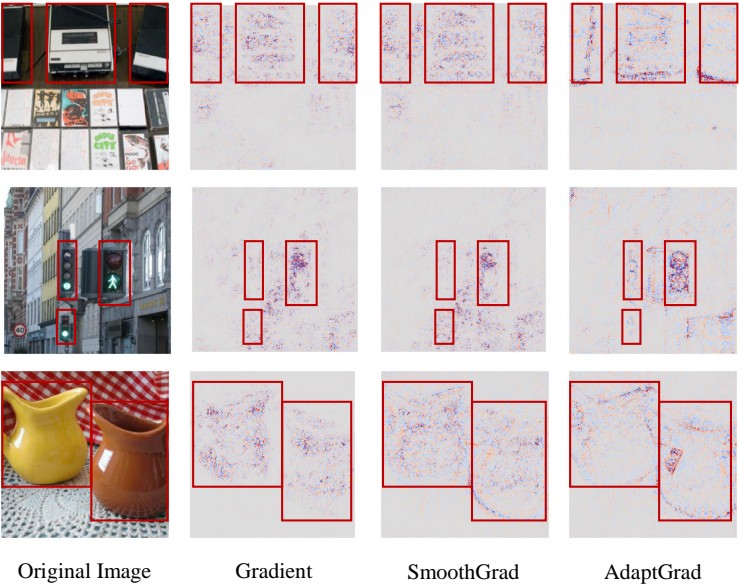

Figure 17: The visual saliency map of multi-object images comparison between Gradient, Smooth-Grad, and AdaptGrad

