# OpenReview forum: "AdaptGrad: Adaptive Sampling to Reduce Noise"
_NeurIPS.cc/2025/Conference — NeurIPS 2025 poster_

### Official Review · Reviewer_CbLr · 2025-06-30

**Clarity:** 3
**Significance:** 1
**Originality:** 2
**Rating:** 3
**Confidence:** 4

**Summary:**

The paper identifies a key limitation in SmoothGrad. SmoothGrad reduces noise in gradient-based saliency maps (e.g., Grad-CAM) by averaging gradients from inputs perturbed with Gaussian noise. However, its fixed noise variance hyperparameter causes out-of-range sampling, introducing extra noise into smoothed gradients. To address this, the authors propose AdaptGrad, an "adaptive" method that dynamically adjusts σ per input dimension. This ensures sampling stays within the valid data domain with a user-defined confidence level.

**Questions:**

Please address the 3 weakness points;

**Ethical Concerns:**

["NO or VERY MINOR ethics concerns only"]

**Final Justification:**

I thank the authors for their reply. My concerns are partially addressed, thus I raised my score from 2 to 3.

I still believe the effectiveness of the work is not clear without comparing to other more general XAI methods in publich benchmarks. The author confine the paper only to "gradient-smoothing" XAI which is a very niche topic that is not significant enough for top conferences like NeurIPS (more like a workshop paper IMHO).

**Limitations:**

yes

**Quality:**

1

**Strengths And Weaknesses:**

Pros:
1. the paper is easy to follow;
2. it identifies some interesting limitations of SmoothGrade
3. it appears to be a practical and versatile XAI tool.

Cons:
1. The evaluation is not sufficent. First, very limited XAI methods are compared. Second, the metrics are insufficient and non-standard. some very common and established faithfulness metrics like Deletion/Insertion score and Pointing Game (for localization accuracy) are missing. Third, it neglects common XAI benchmarks like OpenXAI or Quantus.
2. The proposed method is very incremental to SmoothGrad without broader XAI implications, thus the novelty and significance of the method is very limited. It reads better suited as a workshop paper rather than NeurIPS, IMHO.
3. The table 1 shows very limited correlation.

---

> ### Author Rebuttal · Authors · 2025-07-30
>
> Dear Reviewer,
>
> We greatly appreciate your comments and the effort you put into reviewing our work. We have carefully considered your remarks and will respond according to the following points.
>
> ## The limitations of comparison with other XAI methods
>
> Our comparative analysis of XAI methods focuses exclusively on established gradient-smoothing baselines. We contend that comparing AdaptGrad with non-gradient-smoothing XAI approaches would be inappropriate, as their underlying principles and implementation mechanisms are fundamentally distinct from AdaptGrad’s core framework.
>
> While acknowledging the inherent limitations of our current evaluation scope, we emphasize that the near-universal reliance on gradient information across XAI methodologies renders comprehensive coverage of all XAI techniques practically infeasible within a single study.
>
> Our directly relevant comparisons center on baseline gradient-smoothing works:
>
> - SmoothGrad: The original study limited quantitative comparisons to SmoothGrad and SmoothGrad+Integrated Gradients (IG).
> - NoiseGrad: Evaluated SmoothGrad, NoiseGrad, and their hybrid variant, alongside combinations with Integrated Gradients and Gradient Shap.
>
> Thus, our evaluation framework aligns with established methodological conventions in gradient-smoothing research. We maintain that this approach provides a rigorously controlled basis for assessing AdaptGrad’s contributions relative to its conceptual predecessors. In the revised manuscript, we will elaborate on this rationale and explicitly contextualize our comparative strategy within the broader XAI literature.
>
> ## The limitations of the metrics
>
> We would like to clarify that the Information Level score we employ represents a more quantified metric for AUC/deletion & insertion scores. This methodology was first introduced in [1] and has been widely adopted for evaluating saliency maps in Integrated Gradients-based XAI methods. Specifically, it quantifies deletion scores through an information entropy-based approach, calculating the area under the accuracy-information curve derived from entropy values. This formulation effectively circumvents experimental artifacts arising from "human-selective" deletion and insertion processes. In summary, the Information Level score constitutes an entropy-based refinement of AUC/deletion & insertion scores. Additionally, the adversarial example generation task presented in our appendix serves as a complementary evaluation to these standard metrics.
>
> Regarding common XAI benchmarks, we acknowledge the current limitations in our evaluation scope. However, we wish to clarify that our evaluation framework is primarily derived from Quantus. The Quantus framework encompasses 27 distinct evaluation metrics, rendering its full implementation computationally prohibitive for our research context. This complexity is further exacerbated when integrating AdaptGrad with other gradient-based XAI methods (e.g., NoiseGrad and Integrated Gradients), which significantly amplifies both evaluation intricacy and computational demands. Consequently, we selectively adopted key metrics from Quantus:
>
> - Input Invariance (termed Invariance in our paper),
> - Completeness (termed Consistency in our paper), and
> - Sparseness from the Complexity category (retained as Sparseness in our paper).
> - For localization-related metrics (e.g., Pointing Game and AUC), we utilized the Information Level Score—a more robust and stable alternative—as our primary evaluation tool.
>
> We further note that Quantus lacks efficient mechanisms for controlling computational overhead in such assessments. To address this, we independently designed a dedicated Object Detection task to rigorously evaluate AdaptGrad’s performance.
>
> We recognize that our current evaluation does not incorporate robustness-focused metrics (e.g., those in Quantus’ Robustness category) or benchmarks like OpenXAI. While we acknowledge this as a potential limitation, we maintain that our streamlined evaluation framework—centered on theoretically grounded metrics and task-specific validation—sufficiently comprehensively characterizes AdaptGrad’s capabilities across critical XAI dimensions.
>
> ## The limitations of the novelty and significance
>
> The primary contribution of AdaptGrad lies in proposing a novel gradient smoothing method that effectively eliminates extra noise while demonstrating superior performance across multiple benchmark models. We acknowledge that AdaptGrad builds upon SmoothGrad, but we contend that this advancement constitutes a meaningful innovation. While SmoothGrad achieves notable success in gradient smoothing, it still suffers from residual extra noise. AdaptGrad resolves this issue through the introduction of an adaptive sampling range, thereby significantly enhancing performance across diverse benchmark models.
>
> Critically, we theoretically establish the concept of extra noise for the first time and empirically validate its existence. We assert that this contribution holds substantial significance for the XAI field, as it provides new theoretical foundations and methodological directions for gradient smoothing techniques. Both theoretically and experimentally, our work offers fresh perspectives for XAI research—particularly through the discovery that AdaptGrad can be seamlessly integrated with other gradient-based XAI methods, yielding substantial performance improvements. Consequently, we find it difficult to concur with the characterization of our work as merely "incremental."
>
> ## The correlation between extra noise and noise in saliency maps
>
> What we need to clarify is that the correlations in Table 1 were assessed using a t-test. The Sparseness metric itself does not directly measure the amount of noise. Therefore, the correlation values in Table 1 do not directly indicate the relationship between the additional noise and the noise in the significance map. However, they can indirectly reflect this relationship based on whether they pass the t-test. If the p-value is less than 0.10, it can be inferred that there is a significant correlation between the additional noise and the noise in the significance map.
>
> [1] A. Kapishnikov, T. Bolukbasi, F. Viegas, and M. Terry, “XRAI: Better Attributions Through Regions,” in 2019 IEEE/CVF International Conference on Computer Vision (ICCV), Oct. 2019, pp. 4947–4956. doi: 10.1109/ICCV.2019.00505.

---

> ### Author Response · Authors · 2025-08-07
> **Clarification about faithfulness criteria**
>
> Dear AC,
>
> We gratefully acknowledge your comments. To directly address your valid concerns, we have adopted the standard Deletion/Insertion AUC metrics for a more direct and unambiguous evaluation of faithfulness. We will add these experiments (Table 1 and Table 2, tested on random selected 10 samples) and will update all results in our revised version.
>
> This action also clarifies the ambiguity around our previous "Information Level score," which was conceptually similar but less direct. We will add it with these new, standard metrics throughout the revised manuscript to ensure clarity.
>
> Table 1: Deletion AUC metric results on different models and baselines. (Lower is better)
>
> | Method | VGG16 | InceptionV3 | ResNet50 |
> | -- | --- | --- | --- |
> | VG | 0.0391| 0.1744 | **3.5498** |
> | SG | 0.0392 | 0.1560 | 4.8314 |
> | AG | **0.0347** | **0.1496** | 4.7554 |
>
> Table 2: Insertion AUC metric results on different models and baselines. (Higher is better)
>
> | Method | VGG16 | InceptionV3 | ResNet50 |
> | -- | --- | --- | --- |
> | VG | 0.1214 | 0.3913| 3.8415 |
> | SG | 0.1488 | **0.5130** | **5.3258** |
> | AG | **0.1707** | 0.5010 | 5.1627 |
>
> We will gradually update all the experimental results during the discussion. Additionally, we will provide a detailed description of these results in the revised manuscript.

---

> ### Author Response · Authors · 2025-08-07
> **Clarification about comparison with other XAI methods**
>
> Regarding your second point on baseline comparisons, your feedback has prompted us to think about how to best frame our contribution for a broad audience. We agree a clearer explanation of our methodology is needed.
>
> Our goal in comparison is to demonstrate that **AdaptGrad can replace SmoothGrad as a more general and effective gradient smoothing method**, rather than to prove that AdaptGrad is the optimal solution among all XAI methods. The most rigorous way to validate this is a "controlled substitution" within the same framework.
>
> As an example of the comparisons, we have implemented in our paper: **we aim to show that AdaptGrad + Integrated Gradients > SmoothGrad + Integrated Gradients > Integrated Gradients, rather than AdaptGrad > Integrated Gradients.**
>
> The latter cannot independently verify the improvements we've made to gradient smoothing, because the two approaches differ not only in how they compute gradients, but also in their underlying theories. Comparing against a method from a different family (like LIME, SHAP etc.) would introduce confounding variables, making it impossible to isolate our specific contribution.
>
> While we believe our current experimental design is the most rigorous for our specific scientific question, we recognize that not all readers will be deeply familiar with these subfield-specific conventions.
>
> Therefore, to address your concern, we propose to add a new subsection in Appendix to our paper dedicated to explaining this evaluation methodology. In this new section, we will detail the taxonomy of XAI methods, clarify AdaptGrad's position as a component-level improvement, and explicitly justify our choice of a controlled, comparative evaluation.
>
> We believe this significant addition will fully address the ambiguity you've pointed out, making our paper's contributions clearer and more accessible to a broader audience, without compromising the scientific integrity of our experiments.

---

### Official Review · Reviewer_Dez6 · 2025-07-02

**Clarity:** 3
**Significance:** 3
**Originality:** 2
**Rating:** 5
**Confidence:** 3

**Summary:**

The paper introduces AdaptGrad, a novel adaptive gradient smoothing method designed to reduce noise in gradient-based interpretability techniques such as SmoothGrad. By reinterpreting SmoothGrad as a convolution and analyzing its behavior, the authors identify a key source of “extra noise” caused by out-of-bound sampling. AdaptGrad addresses this by adaptively restricting the sampling range using confidence intervals based on data bounds. The method is theoretically motivated, computationally efficient, and empirically shown to improve saliency map quality across multiple models and interpretability methods.

**Questions:**

1.  How sensitive is AdaptGrad’s performance to the choice of the confidence level, especially in domains with different data distributions or input scales?

2.  Can AdaptGrad be effectively extended to non-vision domains (e.g., NLP or tabular data), where input bounds and semantics differ significantly?

**Ethical Concerns:**

["NO or VERY MINOR ethics concerns only"]

**Final Justification:**

I am convinced of the merits of this work and suggest accepting.

**Quality:**

3

**Strengths And Weaknesses:**

Strengths
-The authors provide a clear theoretical reinterpretation of SmoothGrad using convolution, which leads to identifying a new source of noise (extra noise) and motivates the development of AdaptGrad.
-AdaptGrad is conceptually straightforward, easy to implement, model-agnostic, and computationally efficient—making it broadly usable across existing gradient-based interpretability pipelines.
- The method is evaluated using a wide range of metrics, datasets , models, and explanation methods , including indirect tasks such as object localization and adversarial robustness.

Weaknesses
- The evaluation of noise relies primarily on the Sparseness metric, which is not a direct measure of explanation quality or ground truth relevance, and may not capture all noise-related phenomena.
-The evaluation is conducted on standard image classification tasks and does not demonstrate performance or applicability in more complex or high-stakes real-world domains

---

> ### Author Rebuttal · Authors · 2025-07-30
>
> Dear Reviewer,
>
> We sincerely appreciate your thoughtful feedback and the acknowledgment of our contributions. Your insights are invaluable to us, and we are committed to addressing your comments thoroughly.
>
> ## More evaluation and merits of AdaptGrad
>
> We acknowledge that selecting the right metrics is a challenging and complex task. We opted to rely on the Sparseness metric because it intuitively reflects the clarity of the visualized Saliency Map. Additionally, we referred to other relevant metrics used in related work, such as SmoothGrad and NoiseGrad methods. Furthermore, we introduced two additional evaluations based on visual tasks to further assess the performance of AdaptGrad.
>
> We are grateful for the suggestions you've provided, especially regarding evaluations on real-world data. However, due to the significant workload involved, we are currently limited to assessments based on the ImageNet dataset. In our future work, we will consider conducting evaluations in more complex real-world domains and incorporate those results into the revised version.
>
> ## The sensitivity of AdaptGrad to confidence level
>
> In Appendix A, we discuss the condition of confidence level $c$ in AdaptGrad. Our experiments demonstrate that AdaptGrad's performance remains relatively stable across a range of confidence levels.
>
> However, due to the limited selection of experimental datasets, our current experiments are confined to the ImageNet dataset. This does not fully represent AdaptGrad's performance across other domains with different data distributions and input scales. We plan to expand our work to other datasets in the future and include those results in the revised version.
>
> ## Discussion of broader applications of AdaptGrad
>
> We appreciate your suggestion to extend AdaptGrad to non-visual tasks. Currently, the majority of gradient-based explanation methods focuses almost exclusively on visual tasks. As a result, we haven't yet applied AdaptGrad to non-visual tasks. However, we believe that the concepts and methods behind AdaptGrad can be extended to other domains, especially when dealing with tasks that involve complex structures and high-dimensional inputs. We will explore these possibilities in our future work.

---

> > ### Comment · Reviewer_Dez6 · 2025-08-09
> >
> > I thank the authors for their response. I have no further questions or concerns. I am convinced of the merits of this work and, therefore, leaving the score as is.

---

### Official Review · Reviewer_G7TC · 2025-07-02

**Clarity:** 3
**Significance:** 3
**Originality:** 3
**Rating:** 4
**Confidence:** 4

**Summary:**

This paper studies a problem about generating visual saliency maps with better quality. A novel method named AdaptGrad is proposed to reduce the noise issue in gradient computation and improve the performance for better explanation.

**Questions:**

What’s the difference of statistical behaviour when increasing N? How to decide an optimal N with good explanation quality but minimum computation cost?

The proposed method is pixel level-based computation. While the visual information is usuall redundant, it would be better to run the algorithm with group style.

**Ethical Concerns:**

["NO or VERY MINOR ethics concerns only"]

**Limitations:**

See weakness and question. Authors also provide a discussion section about the limitation at the end of paper.

**Paper Formatting Concerns:**

No.

**Quality:**

3

**Strengths And Weaknesses:**

Strength:

1) The paper is well-organized and easy to follow.

2) The related works are listed in detail with clear background, as well as the pros/cons.

3) The analysis on SmoothGrad is interesting and motivated. The design of per-pixel variance in gradient computation is simple but novel.
Experiment results show the effectiveness of the proposed method. The visual explanation map seems better than baseline methods.

Weakness:

1) Experiments are limited. Only 1000 images from the ImageNet dataset is collected for evaluation. It is unsure about the stability and robustness of the proposed method  against different set of domain of images. No statistic significance is provided.

2) Metrics are limited. Previous work usually use the AUC/deletion&insertion score to evaluate the quality of saliency map, while this work uses the sharpness and information level score, which are rare to use.

3) The confidence is still required to manually tuned for better performance.

---

> ### Author Rebuttal · Authors · 2025-07-30
>
> Dear Reviewer,
>
> We greatly appreciate your constructive comments and the effort you put into reviewing our work. We are pleased that you have acknowledged our contributions and provided such insightful feedback. We have carefully considered your remarks and will respond according to the following points.
>
> ## Limitations of the experiment
>
> We acknowledge that the current experiments are limited to 1,000 images from the ImageNet dataset. At this time, we have not thoroughly tested the stability and robustness of AdaptGrad across different collections of images. We plan to expand our work to other datasets in the future and include the results in the revised version. In our response to reviewer AKq9, we have provided some experimental results that include statistical metrics. Due to the limitations in the efficiency of certain interpretation methods and the calculation of metrics, we are currently only able to provide statistical results for some of the indicators. We will include the complete experimental results in the revised version.
>
> In some related work, such as SmoothGrad, visualization experiments were conducted on only a very small number of samples, and NoiseGrad was tested using just 250 images from ImageNet. In the revised version, we will provide additional experimental results to verify the performance of AdaptGrad on a broader range of data.
>
> ## Limitations of the metrics
>
> We would like to clarify that the information level score we used is a more quantitative metric for AUC/deletion&insertion score, which was first proposed in [1] and is widely used to test the saliency maps of IG methods. This metric quantifies the deletion score as an information entropy-based measure. Based on the entropy value, it calculates the area of the accuracy information curves, thereby obtaining the information level score. This can avoid issues in experimental results caused by "human-selective" deletion and addition. **In short, the information level is an AUC/deletion&insertion score based on information entropy**.
>
> Additionally, the Adversarial Sample Generation Task we designed in the appendix is a supplement to the AUC/deletion&insertion score. We will supplement relevant discussions in the revised version.
>
> ## Manual tuning of confidence $c$
>
> We acknowledge that the confidence level $c$ in AdaptGrad requires manual tuning. However, we have found that the performance of AdaptGrad is relatively stable across a range of confidence levels, as proven by our experiments in Appendix A. In our experiments, we used a confidence level of 0.95, which yielded satisfactory results. We will clarify this point in the revised version and provide more details on how to select the confidence level.
>
> ## Discussion of choice of N
>
> As we mentioned in Section 3.1, increasing N tends to make the Saliency Map converge. Generally speaking, referring to methods like SmoothGrad, NoiseGrad, and Integrated Gradients, we empirically choose N to be 50.
>
> However, we also note that, as shown in Appendix A Table 4, AdaptGrad can achieve good results with a smaller N because it largely eliminates extra noise. We will further discuss the choice of N in the revised version.
>
> ## Other comments
>
> We appreciate your suggestions regarding the group style running algorithm. We will conduct relevant experiments and discuss the potential for this improvement.
>
> [1] A. Kapishnikov, T. Bolukbasi, F. Viegas, and M. Terry, “XRAI: Better Attributions Through Regions,” in 2019 IEEE/CVF International Conference on Computer Vision (ICCV), Oct. 2019, pp. 4947–4956. doi: 10.1109/ICCV.2019.00505.

---

> > ### Author Response · Authors · 2025-08-07
> >
> > Dear Reviewer, we greatly appreciate your constructive comments and the effort you put into reviewing our work. We hope that our responses can address your concerns and we look forward to your further feedback. If you have any additional questions or require further clarification, please do not hesitate to reach out.

---

### Official Review · Reviewer_AKq9 · 2025-07-02

**Clarity:** 1
**Significance:** 2
**Originality:** 2
**Rating:** 4
**Confidence:** 3

**Summary:**

The authors investigate the problem of excessive noise when using the SmoothGrad feature attribution method. In this work, they propose a metric for quantifying the excess noise as well as a novel method AdaptGrad which limits the amount of excess noise based on a given threshold. AdaptGrad extends SmoothGrad by adaptively selecting the smoothing parameter $\sigma$.  Quantitative and Qualitative experiments show that AdaptGrad reduces noise compared to SmoothGrad and other gradient-based attribution methods.

**Questions:**

* How does the confidence interval defined in line 172 relate to AdaptGrad? This seems to be a range of values (which could be [0,0] if x_i = x_max) rather than relating to probability.
* Did you explore simply hard thresholding the samples to be within [x_min, x_max], rather than adaptively reducing $\sigma$?
* What was the statistical test performed in Line 150? Do you have results for all experiment settings?

**Ethical Concerns:**

["NO or VERY MINOR ethics concerns only"]

**Final Justification:**

My original score was 3. During the discussion period, the authors included additional results on clipgrad and statistical error, which addressed many of my original concerns. I still have some concerns regarding notation, given that the reviewers cannot see the revised manuscript. Overall, I think the improvements justify increasing the score to a 4.

**Limitations:**

Limitations are discussed in section 6.

**Quality:**

2

**Strengths And Weaknesses:**

**Strengths:**
* The method empirically seems to improve the sparseness and information level of explanations as compared with taking gradients or smoothgrad.
* The proposed method is fast and simple to implement. It can be used in conjunction with other gradient-based methods.

**Weaknesses:**
* Manuscript could be made more rigorous and has issues with clarity (see "other comments" below).
* While the AdaptGrad saliency maps seem to be more "distinctive" compared with SmoothGrad, it's not clear to me that the explanation is actually better. Since the smoothing parameter $\sigma$ decreases for features that are closer to the boundary of [x_min, x_max] does this not simply highlight the dark / light pixels or pixels close to the boundary values?
* No error bars or measures of variability in Table 2 and 3.
* Limited theoretical contribution.

**Other Comments**
* The connection between SmoothGrad and convolutions / expectations (Section 3.1) is well known and used in previous works (e.g. proposition 1 in [1]). It's not clear if section 3 is intended to be background or part of this paper's contribution -- it would be helpful if this was more clearly delineated.
* The paragraph under equation 6 (lines 116-121) is very confusing. In particular, "the integral in Equation 4 which is not performed over $R^D$ but rather over a bounded domain". However, Equation 4 is an indeed an integral over $R^D$ (or at least, it's not otherwise specified). The next line states that the domain of $G$ is defined over $\Omega$ rather than $R^D$, however, this was not previously established in the paper (and the domain of $F$ is $R^D$).
* In Equation 7, the notation $G(x_i + \epsilon_i; x$ \ $x_i)$ is not defined anywhere and conflicts with the previous definition of $G$.
* Equation 10-12 is undefined when $x_i = x_{min}$ or  $x_{max}$
* The symbol * indicates multiplication in Eq. 2 and convolution in Eq. 4, which is confusing.
* It would be helpful if the metrics (e.g. consistency, invariance, sparseness) were defined in the main text, given that they are mentioned frequently. In addition, the definitions in the supplement could be made more formal. It's difficult to understand what each metric is measuring without referencing the original paper.
* In Eq. 14, I believe the $x_i$ values need to be non-negative and sorted in non-decreasing order?

[1] https://arxiv.org/pdf/2006.06643

---

> ### Author Rebuttal · Authors · 2025-07-30
>
> Dear reviewer:
>
> We greatly appreciate your valuable comments. We are delighted that you have recognized our efforts and provided such insightful feedback. We have carefully considered your remarks and will respond according to the following points.
>
> ## Explanation of effects and contributions
>
> Since neural networks are extremely complex, nonlinear functions, and the input features are highly correlated with each other, SmoothGrad tends to out-of-bound sampling significantly, leading to the extra noise we mentioned in Section 3.2. Therefore, rather than simply highlighting brighter or darker pixels, AdaptGrad carefully **controls the out-of-bound sampling behavior across all pixels**, including those in intermediate positions. As can be clearly observed from the images provided in our paper (we highly recommend zooming in on the saliency maps for visual comparison), AdaptGrad smooths out a significant amount of noise outside the image's main objects and offers a noticeable enhancement over other gradient-based methods.
>
> Therefore, we want to once again highlight **the contribution of extra noise**. We have, for the first time, identified the presence of extra noise in gradient smoothing methods. Methods like SmoothGrad and NoiseGrad, on the other hand, simply introduce perturbations to the inputs and parameters. This not only allows us to design a simple and efficient approach like AdaptGrad, but also assists other research focused on noise reduction in saliency maps.
>
> ## The statistical values in Table 2 and Table 3
>
> We apologize for the oversight in providing complete experimental results. We have now included partial error metrics in Table 2 and Table 3. However, we do apologize that, due to the time constraints of the experiment, we were unable to collect all the statistical metrics in such a short period. The NoiseGrad and Integrate Gradient methods require a significant amount of computational resources to test, so we can only provide partial statistical indicators for now.
>
> We re-conducted 5 separate experiments and calculated the average and standard error. We will continue to gather more experimental data in the coming days and will include the full set of statistical metrics in the final version.
>
> Table 2. Partial results of Consistency and Invariance checks for SmoothGrad(SG)， AdaptGrad(AG) and ClipGrad(CG).
>
> | Method      | Grad             | SG               | AG                | CG               |
> | ----------- | ---------------- | ---------------- | ----------------- | ---------------- |
> | Consistency | 0.02076(0.00028) | 0.01911(0.00014) | 0.020239(0.00026) | 0.01773(0.00013) |
> | Invariance  | 0.3483(0.0002)   | 0.3613(0.0009)   | 0.3484(0.0002)    | 0.3625(0.0008)   |
>
> Table 3. Partial results of Sparseness (SS) evaluation of SmoothGrad(SG), AdaptGrad(AG) and ClipGrad(CG).
>
> | Method | VGG16          | InceptionV3    | ResNet50       |
> | ------ | -------------- | -------------- | -------------- |
> | Grad   | 0.5421(0.0000) | 0.5546(0.0000) | 0.5626(0.0000) |
> | SG     | 0.5334(0.0001) | 0.5421(0.0000) | 0.5580(0.0001) |
> | AG     | 0.5605(0.0002) | 0.5621(0.0001) | 0.5731(0.0000) |
> | CG     | 0.5538(0.0000) | 0.5424(0.0001) | 0.5579(0.0003) |
>
> Here, we need to clarify that the values we reported in the Rebuttal differ from those presented in the paper. This is because we conducted the experiments again. Specifically, the ResNet50 model parameters varied due to differences in the code libraries referenced. Pytorch recently updated the ResNet50 model parameters in the Torchvision library, which has led to different results in our experiments compared to before. We will update these results in the final version.
>
> ## The definition of confidence interval
>
> What we want to explain is that the confidence interval defined in line 172 corresponds to the interval in Equation 12. Theoretically, the confidence interval we should use is $[x_i - x_{min}, x_{max} - x_i]$, but since this interval is **not symmetric with respect to $x_i$**, it creates difficulty in solving for the corresponding sample $\sigma_i$ given the confidence level $c$. This is because our sampling distribution is a normal distribution, which is symmetric. We are unable to obtain an **analytical solution** for the following Equation:
>
> $$
> \int_{x_i - x_{min}}^{x_{max} - x_i} \frac{1}{\sqrt{2\pi}\sigma_i} e^{-\frac{(x_i - x)^2}{2\sigma_i^2}} dx = c
> $$
>
> So, we need to further narrow this interval to $[- \min(|x_i - x_{\text{max}}|, |x_i - x_{\text{min}}|), \min(|x_i - x_{\text{max}}|, |x_i - x_{\text{min}}|)]$, in other words, we choose the "shortest" side as the half-length of our confidence interval. This results in a symmetric confidence interval, allowing us to solve for the desired value. This is essentially the calculation method for $\sigma_i$ in formula 12 in the current AdaptGrad.
>
> ## Discussion of the hard thresholding method
>
> Our goal is to eliminate extra noise, rather than simply limiting the sampling to a fixed range. A simple hard thresholding approach would make smooth convolution kernel is not strictly smooth (The term "smooth" here refers to a function that is smooth in the mathematical sense.), resulting in a significant amount of noise in the saliency maps. In contrast, AdaptGrad, by adaptively adjusting the sampling range, is better at highlighting the features of visual targets.
>
> To verify this, we compared AdaptGrad with the simple hard thresholding method, which we named ClipGrad (marked as CG in experimental results). ClipGrad specifically implements direct clipping of the SmoothGrad sampled values within the range $[x_{min}, x_{max}]$. We supplemented the experimental results of this method on the three baseline models, as shown in Table 2 and Table 3. The results demonstrate that AdaptGrad performs better in eliminating noise and highlighting target features.
>
> ## The statistical test performed in Line 150
>
> The experiment we conducted here is to examine whether the extra noise we proposed correlates with the noise in the saliency map. We used a t-test to assess the correlation between the extra noise and the noise in the saliency map. The experiment was carried out on 1,000 samples, and due to space limitations, we are only presenting the results of the final statistical test along with the corresponding p-value. We have provided all the statistical values in the main text, and the detailed experimental results will be included in the supplementary materials.
>
> ## Clarification of the rigor and clarity of the manuscript
>
> ### The connection between SmoothGrad and convolutions / expectations
>
> We acknowledge that the connection is well known and has been used in previous works. This serves merely as preliminary theory to help readers understand our findings regarding extra noise. We will refine this section and cite relevant literature to clearly define this point.
>
> ### The confusion in the paragraph under Equation 6
>
> We apologize if the explanation here caused any misunderstanding for readers. The integral in Formula 4 refers to the integral associated with the SmoothGrad method, whose domain is $ R $. However, what we intend to highlight is that the "correct" domain of the integral in the SmoothGrad method should actually be a bounded domain. The original domain $ R $ is an "incorrect" setting, which leads to a significant amount of extra noise. We will revise the relevant descriptions to improve clarity and readability.
>
> ### The notation $G(x_i+\sigma_i;x \backslash x_i)$ in Equation 7
>
> We would like to clarify that the notation $G(x_i+\sigma_i;x \backslash x_i)$ in Equation 7 originates from Equation 4. The previous definition was made for all dimensions of $x$, while here we redefine $G$ for a single dimension $x_i$ to simplify subsequent formula derivations. There is no conflict with the previous definition; they simply differ in the dimensions defined.
>
> ### Equation 10-12 being undefined when $x_i=x_{min}$ or $x_{max}$
>
> We would like to clarify that when $ x*i = x*{min} $ or $ x\_{max} $, in equation 12, $ \sigma_i = 0 $, which means we will not sample the feature $ x_i $ in this dimension. This is consistent with our theory of extra noise.
>
> ### The symbol usage of \* in the equations
>
> We have adopted the notation for convolution symbols from the field of mathematics. We regret that this may be unfamiliar to some readers, and we will provide additional explanations and context in the manuscript to clarify its usage.
>
> ### Metrics definition and clarity
>
> We acknowledge that the definitions of metrics in the main text may not be clear enough. Due to space limitations in the main text, we were only able to provide the relevant definitions in the appendix. Thank you for pointing this out. We will include definitions for certain metrics such as consistency, invariance, and sparsity in the main text, and make these definitions more formal in the appendix. We will ensure that readers can understand what each metric specifically measures without having to refer to the original paper.
>
> ### Values $x_i$ in Equation 14
>
> The values of $ x_i $ need to be non-negative (which is typically achieved through normalization) and arranged in non-decreasing order. This is determined by the calculation of the Gini coefficient. We will include a clarification of this point in the paper.

---

> > ### Comment · Reviewer_AKq9 · 2025-08-01
> >
> > I thank the authors for their responses. I appreciate the addition of the standard error metrics and the additional clarifications. The additional ClipGrad results are interesting and help to explain my question about hard thresholding.
> >
> > A few more comments to clarify my concerns:
> >
> > **Confidence Intervals.**
> >
> > My main confusion regarding the confidence intervals is that the term "confidence interval" implies that this is interval s.t. there is some population parameter that should be within the interval with some confidence c (i.e. the traditional definition of a confidence interval). However, this doesn't seem to be the case in this paper -- there is no population parameter to infer. Because of this, calling this interval a "confidence interval" and parameter c the "confidence" could be confusing / misleading. Please feel free to correct me if I’m missing some other connection.
> >
> >
> > **Equations undefined when $x_i = x_{min}$ or  $x_{max}$**
> >
> > I appreciate the clarification. This makes sense in practice when sampling, but please make sure that the notation and equations are fixed in your revision so that they are not undefined.
> >
> >
> > **Statistical Test**
> >
> > Thank you for the details of the test. I want to clarify that I was asking about the test results for other settings of $\alpha$.
> >
> > **Notation**
> >
> > * Regarding function $G$: My concern is that I don't understand what $x_i + \epsilon_i; x$ \ $x_i$ means. What does the semicolon indicate and what does $x$ \ $x_i$ mean? I don't see this defined in the paper -- feel free to point out where this is defined, if I have missed it.
> >
> > * As I was rereading Eq. 7, I noticed that the line below it says "$x_i + \sigma$ is also a random variable", however I think this should be $x_i + \epsilon_i$?.
> >
> > * I have no issues with using the symbol $*$ for convolution -- my point was just that it's overloaded, since it's also used for multiplication in Eq. 2.

---

> > > ### Author Response · Authors · 2025-08-02
> > >
> > > Dear reviewer, we sincerely thank you for your meticulous review and valuable feedback. We appreciate the time and effort you have invested in evaluating our work. Your insights have been instrumental in helping us improve the clarity our manuscript.
> > >
> > > ## Confidence Intervals
> > >
> > > We acknowledge that this definition may cause confusion for readers. The reason we refer to this interval as a confidence interval is that the sampling distribution of AdaptGrad follows a normal distribution, which allows us to compute the sampling standard deviation through the confidence interval. Correspondingly, we term the parameter $c$ the confidence level.
> > >
> > > We have carefully considered your suggestion and will redefine the sampling interval of AdaptGrad. We will adopt a more intuitive definition: the **sampling interval** will be denoted as $[x_{min}, x_{max}]$, where $x_{min}$ and $x_{max}$ represent the minimum and maximum values of the feature values, respectively. This adjustment will make the sampling behavior of AdaptGrad clearer and more comprehensible. Correspondingly, the parameter $c$ will be redefined as the **extra noise level**, further enhancing the intuitiveness and interpretability of AdaptGrad's sampling process.
> > >
> > > ## Equations undefined when $x_i=x_{min}$ or $x_{max}$
> > >
> > > We sincerely appreciate your suggestion. To address the previously undefined part of Equation 12, we revise it as follows:
> > > $$
> > > \sigma_i = \begin{cases}
> > >     \frac{\min\left(x_i - x_{\text{min}} , x_i - x_{\text{max}}
> > > \right)}{\sqrt{2}\text{erfinv}\left(\frac{1 + c}{2}\right)} & \text{if } x_i
> > > \neq x_{\text{min}} \text{ and } x_i
> > > \neq x_{\text{max}}, \\
> > >     0 & \text{if } x_i = x_{\text{min}} \text{ or } x_i = x_{\text{max}}.
> > > \end{cases}
> > > $$
> > > We will ensure that such formulas are clearly defined in the final version, with necessary explanations provided to eliminate any potential confusion.
> > >
> > > ## Statistical Test
> > >
> > > Thank you for your feedback. We have also conducted statistical tests on other settings of $\alpha$. We present the statistical test results for all $\alpha$ values in Table 1.
> > >
> > > Table 1. The statistical test results for different $\alpha$ values.
> > >
> > > |$\alpha$|correlation coefficient|p-value|
> > > |-|-|-|
> > > |0.1|-0.0155|0.6341|
> > > |0.2|-0.0205|0.0961|
> > > |0.3|-0.0261|0.4088|
> > > |0.4|-0.0192|0.5443|
> > > |0.5|-0.0221|0.4840|
> > >
> > >
> > > However, we would like to clarify that the non-significance of the statistical test results for other $\alpha$ ($\alpha\neq 0.2$) values does not imply that our hypothesis is incorrect. Rather, it indicates that under these $\alpha$ values, due to the smaller proportion of additional noise relative to the total noise, we cannot statistically demonstrate the presence of additional noise. This is evidenced by Table 4 in Appendix A, where we observe that the overall noise levels are relatively high when $\alpha$ takes other values (e.g., 0.3, 0.4, 0.5). This aligns with the conclusion drawn in the SmoothGrad paper (Section 3.2). Therefore, in our paper, we selected $\alpha=0.2$ as the primary experimental result to better highlight the presence of additional noise, aiming to prevent readers from being misled by the combination of high noise levels and non-significant test results under other α values.
> > >
> > > If you have any remaining questions on this point, we would be delighted to engage in further discussion. We consider this a particularly intriguing research direction, as it implies the composition of different noise components in Saliency Maps and the patterns of their variation with parameter settings. We aim to delve deeper into this direction in our future work.
> > >
> > > ## Clarification of notation
> > >
> > > We apologize for the confusion caused by the notation in our paper.
> > >
> > > First, regarding the notation $x_i + \varepsilon_i; \mathbf{x} \backslash x_i$ in Equation 7, we wish to clarify that it indicates we perturb only $x_i$ while keeping all other features in $\mathbf{x}$ unchanged. We use the set-theoretic notation $\mathbf{x} \backslash x_i$ to denote all features of $\mathbf{x}$ except $x_i$, and the semicolon $;$ to separate the variables from the parameters within the function. For example, $f(x; y)$ signifies that $x$ is the input variable and $y$ is the parameter of function $f$. We will add further explanations in the footnote of Equation 7 to help readers better understand this notation.
> > >
> > > Second, we must acknowledge an error in the notation used in Line 132 below Equation 7. There, we incorrectly represented the random variable $x_i + \varepsilon_i$ using the symbol $x_i + \sigma_i$. We will correct this mistake and ensure consistent notation throughout the paper.
> > >
> > > Lastly, regarding the use of the convolution symbol $\ast$, we will clarify its distinction from multiplication: we will use $\cdot$ to denote multiplication and $\ast$ exclusively for convolution. We will also provide additional explanations in the paper to enhance readers' understanding of these notations.

---

> > > > ### Comment · Reviewer_AKq9 · 2025-08-04
> > > >
> > > > I thank the authors for their additional clarifications. I still have some reservations about notation, especially since the reviewers can't see the revised manuscript. However, given the proposed changes regarding manuscript notation and clarity, and the additional results on clipgrad and statistical error, I will increase my score to 4.

---

> > > > > ### Author Response · Authors · 2025-08-07
> > > > >
> > > > > Dear reviewer, we sincerely appreciate your thorough review and insightful comments. We are grateful for the time and effort you have dedicated to evaluating our work. Your feedback has been invaluable in helping us improve the clarity and quality of our manuscript. We will ensure that all your suggestions are carefully considered and incorporated into the revised version of our paper.

---

### Decision · Program_Chairs · 2025-09-17

**Decision:**

Accept (poster)

**Comment:**

The paper proposes AdaptGrad, an improved version of SmoothGrad, where the noise level is adapted for each input dimension.  Experimental results show that AdaptGrad significantly improves the explainability of SmoothGrad.

Reviewers acknowledged significant contributions in improving the gradient-based explanation methods with a simple idea, but also raised several concerns, including clarity issues, limited experiment, missing criteria, and missing baselines.  The authors' rebuttal addressed those concerns with promises to include additional experimental results.

This is a borderline paper, and the reviewer-AC discussion concluded that, although the scope of the paper is niche, the proposed method is useful for many people who use gradient-based explanation methods.  The main experiment comparing the proposed method with the other gradient-based methods is fine, but the performance comparison with other types of sota explanation methods is also important.  The proposed method may not be sota in terms of the explanation quality.  Then, the authors should discuss why the proposed method is still important and useful.